# Cell detoxification of secondary metabolites by P4-ATPase-mediated vesicle transport

Yujie Li[1†], Hui Ren[1†], Fanlong Wang[1], Jianjun Chen[1], Lian Ma[1], Yang Chen[1], Xianbi Li[1], Yanhua Fan[1], Dan Jin[1], Lei Hou[1], Yonghong Zhou[1], Nemat O Keyhani[2], Yan Pei[1]*

[1]Biotechnology Research Center, Southwest University, Chongqing, China; [2]Department of Microbiology and Cell Science, Institute of Food and Agricultural Sciences, University of Florida, Gainesville, United States

**Abstract** Mechanisms for cellular detoxification of drug compounds are of significant interest in human health. Cyclosporine A (CsA) and tacrolimus (FK506) are widely known antifungal and immunosuppressive microbial natural products. However, both compounds can result in significant side effects when used as immunosuppressants. The insect pathogenic fungus *Beauveria bassiana* shows resistance to CsA and FK506. However, the mechanisms underlying the resistance have remained unknown. Here, we identify a P4-ATPase gene, *BbCRPA*, from the fungus, which confers resistance *via* a unique vesicle mediated transport pathway that targets the compounds into detoxifying vacuoles. Interestingly, the expression of *BbCRPA* in plants promotes resistance to the phytopathogenic fungus *Verticillium dahliae* via detoxification of the mycotoxin cinnamyl acetate using a similar pathway. Our data reveal a new function for a subclass of P4-ATPases in cell detoxification. The P4-ATPases conferred cross-species resistance can be exploited for plant disease control and human health protection.

**\*For correspondence:** peiyan3@swu.edu.cn

[†]These authors contributed equally to this work

## Editor's evaluation

This important study reveals a new strategy for protecting plants from certain fungal pathogens. The authors show convincing data for a mechanism of resistance that involves sequestration of a fungal toxin into vacuoles. This work will be of interest to mycologists and scientists interested in plant biotechnology.

## Introduction

The mining of bioactive molecules known as secondary metabolites or natural products is of significant interest relevant to almost all aspects of human activity from food security to health and well-being. Microbes are known reservoirs for the production of such natural products, that has included the discovery of the world's first broad-spectrum antibiotic from *Penicillium notatum* to a range of diverse chemical compounds that continue to be characterized from fungi today (*Brakhage, 2013*; *Chanda et al., 2009*; *Fleming, 1944*; *Keller et al., 2005*). Two of the most widely known microbial products of human health relevance are cyclosporine A (CsA), a neutral lipophilic cyclic polypeptide originally isolated from the entomopathogenic fungus *Beauveria nivea* (*Tolypocladium inflatum*) and FK506 (tacrolimus), a macrolide lactone isolated from *Streptomyces tsukubaensis* (*Gupta et al., 1989*; *Odom et al., 1997*; *Tanaka et al., 1987*). Although both of these molecules were originally isolated due to their antifungal properties, their functioning as immunosuppressive agents has revolutionized

aspects of medicine (*Borel et al., 1976*; *Beauchesne et al., 2007*; *Dreyfuss et al., 1976*; *Guada et al., 2016*; *Margaritis and Chahal, 1989*; *Odom et al., 1997*; *Tanaka et al., 1987*). The major mechanism mediating the antimicrobial activities of both CsA and FK506 appear to be *via* inhibition of calcium signaling by targeting of calcineurin (CaN) through cyclophilin A-CsA and FKBP-FK506 complexes, respectively (*Liu et al., 1991*; *Sharma et al., 1994*). In terms of human health relevance, however, CsA and FK506 are highly used immunosuppressants that have been applied in a wide variety of therapeutic applications from facilitating human organ transplants to autoimmune-disease therapies, hypertension, and even ocular diseases (*Borel and Gunn, 1986*; *Thomson et al., 1993*; *Tory et al., 2008*). However, both compounds can also result in significant side effects that can include nephro- and hepatotoxicities, central nervous system (CNS) disturbances, hirsutism, and gingival hyperplasia (*DiMartini et al., 1996*; *Kaeberlein, 2013*). Little, however, is known concerning mechanisms for detoxification of these drugs, and to date no pathways have been identified for mediating (microbial or other) resistance (s) to these compounds. Resistance can be of further importance due to the effects of similar fungal secondary metabolite toxins produced by phytopathogens that cause significant agricultural damage and decreased productivity. Within this context the phyotopathogenic fungi, *Verticillium dahliae Kleb* and *Fusarium graminearum*, causing Verticillium wilt corn ear rot and wheat head blight disease, respectively, are two of the most destructive diseases of many important crops (*Subbarao et al., 1996*; *Sutton, 1982*; *Veronese et al., 2003*), and mycotoxins, such as trichothecenes produced by *F. graminearum*, present in foods and forages cause serious risks to the health of animals and human beings (*Berthiller et al., 2013*; *D'Mello et al., 1999*).

Most organisms are endowed with two major mechanisms for detoxification of small molecular weight chemical compounds: (i) chemical modification (s) resulting in inactivation, which can include hydrolysis and/or oxidation, and conjugation, (ii) compartmentation, and eventual degradation (*Berthiller et al., 2013*; *Coleman et al., 1997*). In compartmental detoxification, molecules (toxins) are transported into structures (organelles) where they are sequestered and degraded. As part of these processes, it had been demonstrated that vesicle-mediated transport can contribute to secondary metabolite sequestering in order to protect the host cell (or resistant organisms) from self-toxicity (*Sirikantaramas et al., 2008*). In fungi and plants, the primary subcellular compartment for detoxification is the vacuole, while in animals it is the lysosome. It is well known that ATP-binding cassette (ABC) transporters can catalyze drug/toxin efflux across membranes out of cells and/or into specialized compartments as part of mechanisms involved in drug resistance and detoxification (*Coleman et al., 1997*; *Theodoulou, 2000*; *Sipos and Kuchler, 2006*; *Wolfger et al., 2001*). Most lipid ABC transporters are floppases mediating the movement of phospholipids from the cytosolic surface to the extracellular leaflet (*Coleman et al., 2013*; *Perez et al., 2015*; *van Meer et al., 2006*; *Zhou and Graham, 2009*). Unlike ABC transporters, type IV P-type ATPases (P4-ATPases) have been proposed to function as phospholipid flippases that pump specific phospholipid substrates in the reverse direction: from the exofacial to the cytosolic leaflet of membranes (*Coleman et al., 2013*; *Hankins et al., 2015*; *Zhou and Graham, 2009*). P4-ATPases, identified only in eukaryotic cells, constitute the largest subfamily of P-type ATPase, and have important roles in the initiation of the vesicle formation and membrane trafficking by the generation of phospholipid asymmetry in biological membranes, which are involved in a variety of physiological processes, including cell surface growth, the biogenesis of cellular organelles, endocytosis, and protein storage and protein sorting (*De Matteis et al., 2013*; *Hara-Nishimura et al., 1998*; *Lopez-Marques et al., 2014*; *McMahon and Gallop, 2005*; *Rothman and Wieland, 1996*; *van der Mark et al., 2013*). However, to date, the function of P4-ATPases in cellular detoxification of small peptides and/or secondary metabolites has been rarely reported.

It was observed that the insect pathogenic fungus *Beauveria bassiana* exhibits resistance to CsA (*Zhou et al., 2016*). However, the precise mechanisms underlying this resistance have yet to be fully understood. In this study, from a screen of a fungal random insertion mutant library, a *B. bassiana* CsA susceptible mutant was identified. The mutation insertion site was mapped to an open-reading frame coding for a P4-ATPase and designated as BbCrpa (cyclosporine A resistance P4-ATPase). The mechanism for BbCrpa functioning is shown to be via delivery of the toxins into vacuoles through a P4-ATPase-mediated vesicle transport pathway. Interestingly, the expression of *BbCRPA* in *Arabidopsis thaliana* and *Gossypium hirsutum Linn*. (cotton) significantly increased the resistance of transgenic plants against *V. dahliae* toxin, and reduced the severity of Verticillium wilt disease, indicating the utilization of the P4-ATPase endowed detoxification in other species.

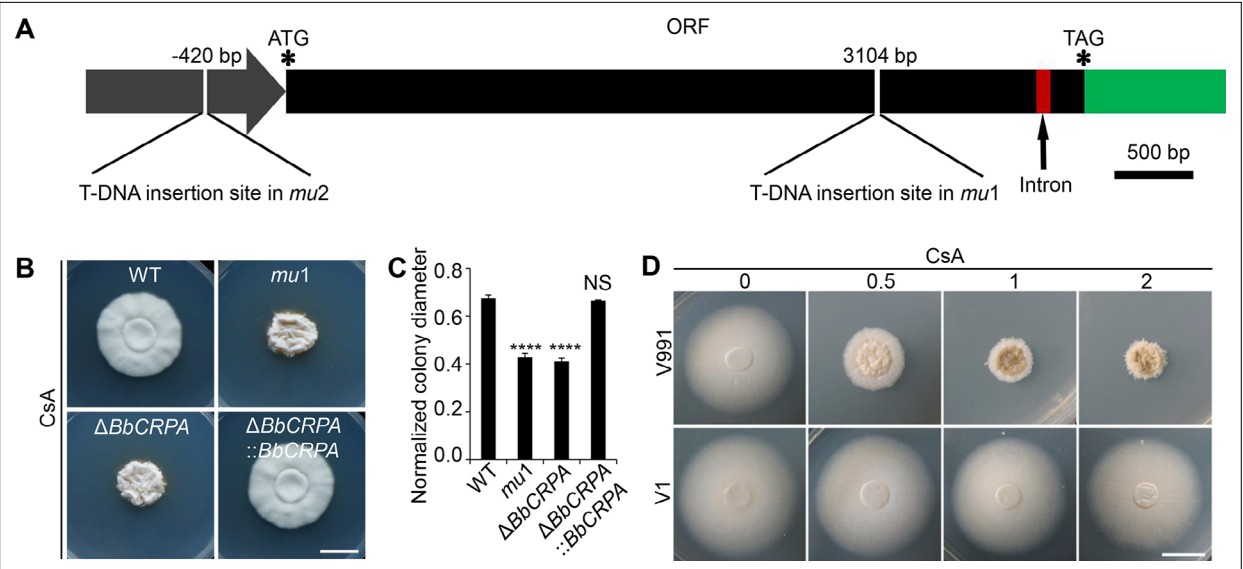

**Figure 1.** BbCrpa confers the resistance of *Beauveria bassiana* and *Verticillium dahliae* to CsA. (**A**) Schematic diagram of T-DNA insertion in *mu*1 and *mu*2. The two T-DNA insertions took place in same gene at different regions: in *mu*1, the insertion site located in the coding region (3104 bp); in *mu*2, the site was in the promoter region (–420 bp). The gene codes for a putative phospholipid-translocating P-type ATPase (P4-ATPase), named BbCrpa (<u>c</u>yclosporine <u>r</u>esistance <u>P</u>4-<u>A</u>TPase) that has a 4080 bp ORF and an intron (73 bp) near its 3' end. (**B and C**) Disruption of *BbCRPA* makes *B. bassiana* sensitive to CsA. The wild-type, *mu*1, *BbCRPA* gene-knockout (Δ*BbCRPA*), and complemented (Δ*BbCRPA::BbCRPA*) strains were grown on CZP +CsA (20 µg/ml). (**D**) Ectopic expression of *BbCRPA* in *V. dahliae* increases the resistance of CsA. Wild-type *V. dahliae* (V991) and V1 (expressing *BbCRPA* in *V. dahliae* V991) were grown on PDA and PDA +CsA (0.5 µg/ml, 1 µg/ml, and 2 µg/ml). For CsA sensitivity analysis, plates were spot inoculated with 3 µl conidial suspensions (1×10⁷ conidia/ml) and incubated at 26°C for about 10 days. The variation in growth rates was shown as [colony diameter CZP supplemented with CsA]/[colony diameter CZP]. All experiments were performed in triplicate (n = 3). Data are represented as the mean ± SD. ****p<0.0001 from Student's *t*-test. NS, not significant. Scale bars, 1 cm for (**B and D**).

The online version of this article includes the following source data and figure supplement(s) for figure 1:

**Source data 1.** Growth of the target strains on CZP supplemented with CsA normalized to growth on CZP.

**Figure supplement 1.** *B. bassiana* shows resistance to CsA and FK506 and identification of CsA-sensitivity mutants, *mu*1 and *mu*2.

**Figure supplement 1—source data 1.** Growth of wild-type stran, *mu*1 and *mu*2 at CZP supplemented with CsA normalized to growth at CZP.

**Figure supplement 2.** Construction and Identification of *BbCRPA* disruption strain.

**Figure supplement 2—source data 1.** qRT-PCR experiments, conidial yield, and growth of target strains at CZP supplemented with CsA normalized to growth at CZP.

**Figure supplement 2—source data 2.** PCR analysis of *BbCRPA* disruption mutant.

**Figure supplement 3.** Distribution of phosphatidylserine (PS) and phosphatidylethanolamine (PE) in the wild-type and Δ*BbCRPA*.

## Results

### BbCrpa is a member of the P4-ATPase subfamily and contributes to CsA and FK506 resistance

CsA is toxic to a number of filamentous fungi, however, some fungi, including the insect pathogen, *Beauveria bassiana*, possesses intrinsic resistance to CsA (*Dreyfuss et al., 1976*; *Traber and Dreyfuss, 1996*; *Zhou et al., 2016*; *Figure 1—figure supplement 1A*). Although structurally different from CsA, the secondary metabolite, FK506, produced by *Streptomyces tsukubaensis*, also displays antifungal activity. Likewise, *B. bassiana* shows resistance to the drug (*Figure 1—figure supplement 1B*).

To uncover the underlying mechanism of the CsA resistance in *B. bassiana*, a random T-DNA insertion mutagenesis library was screened for sensitivity to CsA. From a screen of ~20,000 mutant colonies, two CsA-sensitive mutants, named *mu*1 and *mu*2 were isolated (*Figure 1—figure supplement 1C and D*). Mapping of the T-DNA insertion sites by Y-shaped adapter-dependent extension (YADE) revealed that both mutants had insertions in the same gene, but at different positions (*Figure 1A*, *Figure 1—figure supplement 1E*). The mutant *mu*1 contained an insertion in the coding region (at 3104 bp from the translation start site), whereas the insertion site in *mu*2 occurred in the upstream promoter

sequences (at –420 bp; *Figure 1A*). Bioinformatic analyses of the open reading frame (ORF) indicated that it encoded for a predicted protein with 1359 amino acids. Cluster analysis indicated that the protein belongs to the Type IV P-type ATPase subfamily (P4-ATPases) (*Figure 1—figure supplement 1F and G*). The predicted topological model showed that the protein contains an A-domain (actuator domain), a N-domain (nucleotide binding domain), a P-domain (phosphorylation domain) and 10 predicted transmembrane-spanning segments, which a typical P-type ATPase has (*Figure 1—figure supplement 1H*). Thus, the protein was named as BbCrpa (cyclosporine A resistance P4-ATPase). In order to recapitulate the phenotype, a targeted *BbCRPA* gene-knockout strain was constructed as detailed in the methods section (*Figure 1—figure supplement 2A–C*). The ΔBbCRPA strain exhibited sensitivity to CsA, and the CsA resistance defect was successfully restored by introducing the ectopic expression of *BbCRPA* into the ΔBbCRPA background, indicating that the absence of BbCRPA led to a loss of CsA resistance (*Figure 1B and C*). The ΔBbCRPA strain also became sensitive to FK506, and the FK506-resistance defect could be complemented by the ectopic expression of *BbCRPA* in the ΔBbCRPA background too (*Figure 1—figure supplement 2D and E*). The ΔBbCRPA strain showed an approximate 4.73 ± 0.04% reduction in growth on CZP medium and a decrease in virulence using *Galleria mellonella* larvae as the target host in topical insect bioassays: $LT_{50}$ values for the wild-type=93.13 ± 2.49 hr and for ΔBbCRPA = 103.59 ± 1.28 hr (*Figure 1—figure supplement 2F, G and J*). However, the mutation had no significant impact on sporulation (*Figure 1—figure supplement 2H,I*). To further confirm the function of BbCrpa in CsA resistance, we expressed *BbCRPA* in a phytopathogen strain, *Verticillium dahliae* V991 (the naturally CsA-sensitive fungus compared to *B. bassiana*). The ectopic expression of *BbCRPA* in *V. dahliae* significantly increased fungal resistance to CsA (*Figure 1D*).

The phosphorylation of aspartic acid (D) 614 in the identified conserved P-domain DKTG sequence is crucial for ATPase activity of P-type ATPases (*Palmgren and Nissen, 2011*). In order to examine whether ATPase activity was required for the phenotype of CsA resistance, aspartic acid (D) 614 was replaced with arginine (R). The D614R mutant was no longer able to confer resistance to CsA (*Figure 1—figure supplement 2K and L*). In addition, site directed mutagenesis of a conserved isoleucine (I) located in transmembrane segment M4 of P4-ATPases, to a glutamic acid (as found in Na⁺/ K⁺ pumps) resulted in loss of ability of the protein to confer CsA resistance (*Panatala et al., 2015*; *Figure 1—figure supplement 2K and L*). Indeed, P4-ATPases are known as flippases that catalyze the trafficking of aminophospholipids, for example phosphatidylethanolamine (PE) and phosphatidylserine (PS), from the plasma membrane to the inner membrane system. To determine the flippase activity of BbCrpa, we compared the distribution of PE and PS between wild-type *B. bassiana* and ΔBbCRPA. PE was mainly distributed on the cytosolic face in the wild-type, while no significant PE signal was found in the cytosol of ΔBbCRPA (*Figure 1—figure supplement 3A*). To observe the distribution of PS, we expressed the PS indicator *eGFP::Lact-C2* (*Schultzhaus et al., 2015*) in the cells. In the wild-type cells, PS was enriched mainly in the cytosol, while there was a significant accumulation of PS on the plasma membrane in ΔBbCRPA cells (*Figure 1—figure supplement 3B*). The results indicated hat the trafficking of PE and PS from plasma membrane to the inner membrane system was interrupted when BbCrpa was defeted. Taken together, these data suggests that PE and PS are substrates of BbCrpa.

Although BbCrpa showed about 60% identity to Drs2p, a well-investigated P4-ATPase in *Saccharomyces cerevisiae* (NCBI Gen_locus ID: NP_009376), ectopic expression of *DRS2* in the ΔBbCRPA strain did not restore the resistance to CsA (*Figure 1—figure supplement 2M and N*), suggesting that BbCrpa is functionally different with Drs2p in terms of toxin resistance.

## CsA/FK506 is transported from TGN-EE-LE to vacuoles

As P4-ATPases have been implicated in vesicle formation and trafficking (*Hua et al., 2002*; *Pomorski et al., 2003*; *Poulsen et al., 2008*; *Zhou and Graham, 2009*), we sought to test the hypothesis that CsA/FK506 detoxification may be mediated through a P4-ATPase-mediated vesicle transport process. To this end, a dual labeling system using CsA and FK506 labeled with 5-carboxyfluorescein (5-FAM) and target proteins fused with eGFP/mRFP was employed. The labeling did not significantly affect the toxic activity of the drugs (*Figure 2—figure supplement 1A–D*). To assess a possible interference of 5-FAM and fluorescent proteins cleaved from the fused materials with the distribution assay, we compared the distribution of 5-FAM or eGFP/mRFP alone with that of fused materials. No such

interference was observed: the distribution of eGFP/mRFP or 5-FAM alone differ from that of eGFP/mRFP fused proteins or 5-FAM labeled CsA (*Figure 2—figure supplement 2A–F*). Cells harboring red fluorescent fusion proteins of either the Rab5 or Rab7 GTPase, or the pleckstrin homology domain of the human oxysterol binding protein (PH$^{OSBP}$) were used. The mRFP::PH$^{OSBP}$, mRFP::Rab5, and mRFP::Rab7 fluorescent proteins were used as markers to visualize the location of *trans*-Golgi network (TGN), early endosomes (EEs), and late endosomes (LEs), respectively (*Molinari et al., 1997*; *Pantazopoulou and Peñalva, 2009*; *Sugimoto et al., 2001*). In separate experiments, cells treated with CsA-5-FAM or FK506-5-FAM were co-stained with the membrane-binding fluorescent dye FM4-64 used to label vesicles/vacuoles (*Lewis et al., 2009*). CsA-5-FAM treated wild-type cells showed membrane staining and subsequent accumulation of labeled compound in vesicles (*Figure 2A*) and vacuoles (*Figure 2B and G*), as identified by the FM4-64 co-staining. In contrast to the wild-type cells, almost no CsA-5-FAM signal could be seen inside vacuoles of the Δ*BbCRPA* mutant (*Figure 2C and G*). In the wild-type cells, CsA-5-FAM also co-localized to mRFP::PH$^{OSBP}$-labeled *trans*-Golgi regions, as well as mRFP::Rab5 and mRFP::Rab7-labeled endosomes (*Figure 2D–F*). Similarly, FK506-5-FAM was seen in vacuoles, *via trans*-Golgi/endosome localizations (*Figure 2H–M*).

## BbCrpa acts as a component involved in vesicle trafficking through the *trans*-Golgi-endosomes to vacuoles

A series of ten different eGFP-BbCrpa fusion proteins were constructed to observe the subcellular localization of BbCrpa. By testing for complementation of the Δ*BbCRPA* phenotype, BbCrpa bearing eGFP at the N- or C-terminus, or at the regions between transmembrane segments 4–5, 8–9, and 9–10, remained capable of mediating CsA detoxification (*Figure 3—figure supplement 1A–G*). The eGFP-BbCrpa N-terminal fusion protein (eGFP::BbCrpa) was used for further study. In conjunction with FM4-64 staining, fluorescent signal derived from eGFP::BbCrpa could be seen in the apical plasma membrane and cytosolic structures in germinating conidia and the eGFP signal also emerged in the subapical region called the Spitzenkörper in germ tubes (*Figure 3—figure supplement 1H1*). In hypha, dual labeling with eGFP::BbCrpa and mRFP::PH$^{OSBP}$ revealed the co-localization of both signals in trans-Golgi (*Figure 3A*), with continued co-localization at discrete spots as well as accumulation of the eGFP signal in vacuoles (*Figure 3B*). A similar pattern to mRFP::PH$^{OSBP}$ could be seen with co-staining experiments using FM4-64, which also allowed for staining of the vesicle/vacuole membranes containing the eGFP::BbCrpa signal (*Figure 3C*). Dual labeling with eGFP::BbCrpa and mRFP::BbRab5 (*Figure 3D*), and eGFP::BbCrpa and mRFP::BbRab7 (*Figure 3E*) revealed the localization of BbCrpa in EEs and LEs, respectively. eGFP::BbCrpa and FM4-64 staining showed the localization of BbCrpa in vesicles (*Figure 3C*) or vacuoles (*Figure 3F*). FM1-43 is another membrane probe that has been widely used for monitoring recycling of vesicles (*Hansen et al., 2009*). The staining of FM1-43 revealed that both mRFP::BbRab5 and mRFP::BbRab7 were co-localized with FM1-43 (*Figure 3—figure supplement 1J and K*), and BbCrpa exhibited vesicular and vacuolar localization (*Figure 3—figure supplement 1L*). Feeding 5-FAM-labeled CsA or FK506 to cells expressing an mRFP tagged version of BbCrpa (mRFP::BbCrpa), the two signals were colocalized in some puncta on the plasma membrane, and the signals were ultimately converged in vacuoles (*Figure 3G–J*).

Time-lapse dual-label microscopy combining either eGFP::BbCrpa and FM4-64 or eGFP::BbCrpa and mRFP::PH$^{OSBP}$ indicated the dynamic trafficking through the described pathway from the vesicles to early/late endosome, and to vacuoles (*Figure 3K and L*, *Figure 3—videos 1 and 2*). Whereas the eGFP::BbCrpa and FM4-64 staining showing vacuolar localization of the *B. bassiana* P4-ATPase (*Figure 3F*), the heterologous expression of an eGFP-tagged version of the yeast homolog, eGFP::Drs2p, showed cytoplasmic localization of this protein (*Figure 3M*). Brefeldin A, a fungal metabolite that disrupts Golgi structure and inhibits Golgi function, was introduced to further confirm the movement of BbCrpa from Golgi to vacuole. The result showed that BFA treatment affectes BbCrpa accumulation in the vacuoles (*Figure 3—figure supplement 2*), showing that the trafficking of BbCrpa to vacuoles would be blocked when the function of Golgi (including TGN) is disrupted.

## Contributions of BbCrpa N- and C-terminal tails to CsA/FK506 detoxification

BbCrpa has a 268 amino acid cytosolic N-terminal tail before the first transmembrane segment and an 86 amino acid cytosolic C-terminal tail after the last membrane segment (*Figure 1—figure*

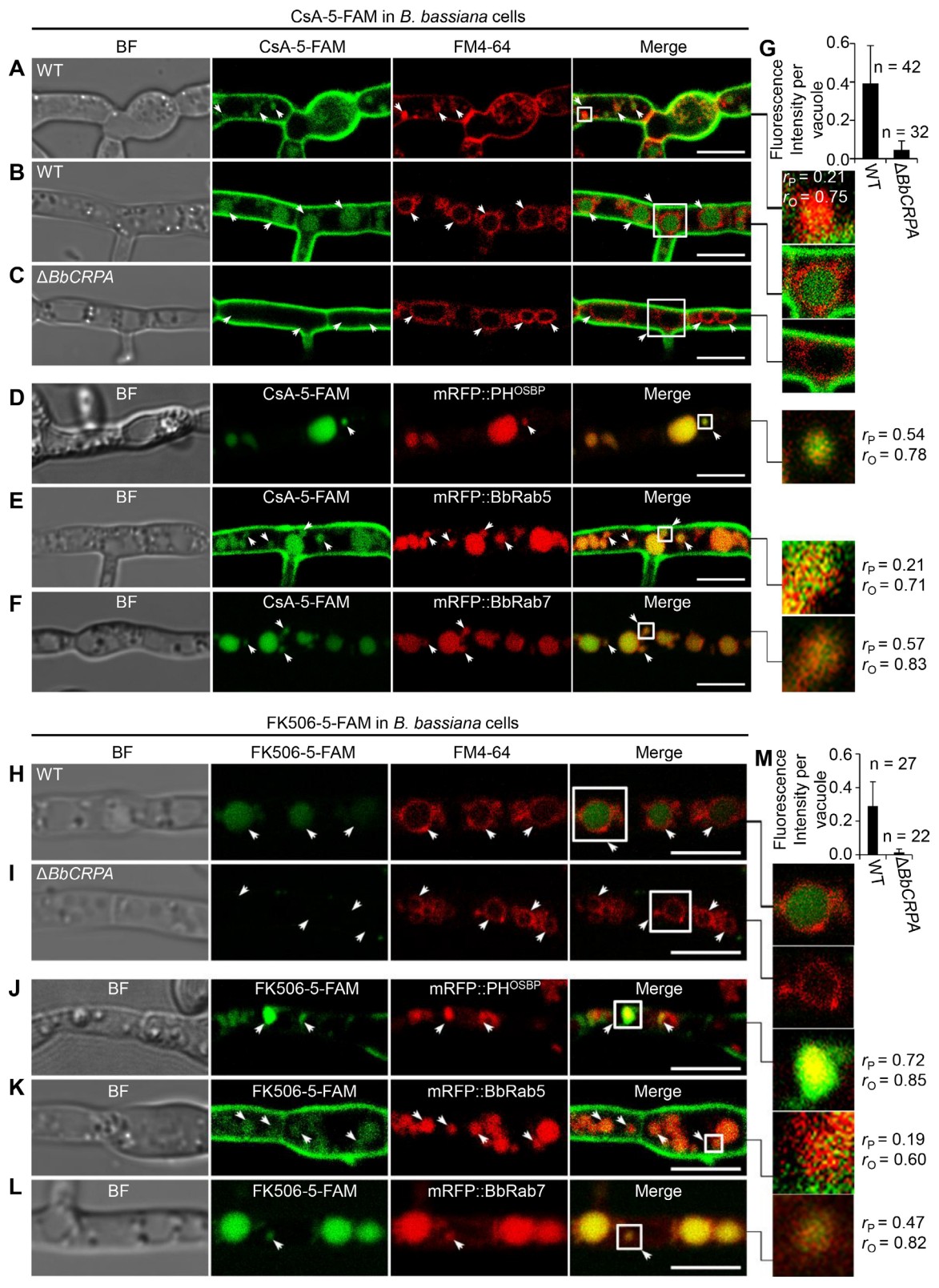

**Figure 2.** Distribution of fluorescein-labeled CsA and FK506 in the wild-type and ΔBbCRPA cells. (A–G) Distribution of fluorescein-labeled CsA. In the wild-type cells, fluorescein-labeled CsA (CsA-5-FAM) appeared in vesicles/endosomes (arrows, stained by FM4-64) (**A**), TGN (arrows, marked by mRFP::PH$^{OSBP}$) (**D**), EEs (arrows, marked by mRFP::BbRab5) (**E**), and LEs (arrows, marked by mRFP::BbRab7) (**F**), and accumulated in vacuoles (arrows, stained by FM4-64) (**B**); while in ΔBbCRPA cells, the fluorescein signal was nearly undetectable in the vacuoles (arrows) (**C**). The fluorescent intensity

*Figure 2 continued on next page*

*Figure 2 continued*

within the wild-type (**B**) and Δ*BbCRPA* (**C**) cells was measured by ImageJ (**G**). (**H–M**) Distribution of fluorescein-labeled FK506. In the wild-type cells, FK506-5-FAM appeared in TGN (**J**, arrows), EEs (**K**, arrows), and LEs (**L**, arrows), and accumulated in vacuoles (**H**, arrows); while in Δ*BbCRPA* cells, the fluorescein signal was nearly undetectable in the vacuoles (**I**, arrows). The fluorescent intensity within the wide-type and Δ*BbCRPA* cells was measured by ImageJ (**M**). The value of Pearson ($r_P$) and Overlap ($r_O$) correlation coefficient shows the extent of colocalization between the two target molecules. The values range between +1 (positive correlation) and –1 (negative correlation). Data are represented as the mean ± SD for (**G and M**). Scale bars, 5 μm for (**A–F, H–L**).

The online version of this article includes the following source data and figure supplement(s) for figure 2:

**Source data 1.** CsA-5-FAM/FK506-5-FAM fluorescent intensity within the wild-type and Δ*BbCRPA* cells.

**Figure supplement 1.** Fluorescent-labeled CsA and FK506 maintain their toxic activity.

**Figure supplement 1—source data 1.** Growth of the target strains at CZP supplemented with 5-FAM/CsA/FK506/CsA-5-FAM/FK506-5-FAM normalized to growth at CZP.

**Figure supplement 2.** Distribution comparison of eGFP/MRFP and 5-FAM alone with that of eGFP/mRFP::BbCrpa and CsA-5-FAM.

*supplement 1H*). Deletion of either the N- (1–268 aa) or the C-terminus (1274–1359 aa) eliminated the detoxification activity of BbCrpa (*Figure 4A–C*, *Figure 4—figure supplement 1A and B*). Microscopic visualization of N-terminal eGFP fusion constructs to the N- and C-terminal deletion mutants (eGFP::dB$^N$ and eGFP::dB$^C$) revealed that loss of the N terminus (eGFP::dB$^N$) eliminated vacuolar localization of the protein, whereas the eGFP::dB$^C$ protein was still able to traffic to vacuoles (*Figure 4—figure supplement 1E*).

In order to further probe the contributions of the N- and C-terminal domains of BbCrpa in mediating CsA/FK506 detoxification, we substituted the homologous N- and C-terminal domains of the yeast Drs2p (that does not complement the CsA/FK506 susceptibility phenotype of the *BbCRPA* mutant strain; *Figure 4A*). The resulting chimeric proteins with either C-terminus or N-terminus from BbCrpa alone unable to restore CsA-/FK506-tolerance to the cell (*Figure 4D and E*, *Figure 4—figure supplement 1C and D*). However, the simultaneous substitution of the BbCrpa N- and C-terminal domains into the yeast protein led to a chimera that showed a significant increase in resistance to CsA/FK506 compared to the natural Drs2p (*Figure 4D and E*, *Figure 4—figure supplement 1C and D*).

To identify the key motif (s) in the N-terminus responsible for vacuolar targeting, a series of N-terminal deletion mutants fused with mRFP were generated. These included: (1) B$^{N1-268}$::mRFP, (2) B$^{N151-268}$::mRFP, (3) B$^{N216-268}$::mRFP, (4) B$^{N226-268}$::mRFP, (5) B$^{N236-268}$::mRFP, (6) B$^{N246-268}$::mRFP, (7) B$^{N256-268}$::mRFP, (8) B$^{N257-268}$::mRFP, (9) B$^{N258-268}$::mRFP, (10) B$^{N259-268}$::mRFP (*Figure 4F*, *Figure 4—figure supplement 1F*). All the mutants retained the wild-type vacuolar localization except B$^{N259-268}$::mRFP (*Figure 4F*, *Figure 4—figure supplement 1F*). These data indicated that the N-terminal sequence, F$^{258}$ASFLPKFLFE$^{268}$, is critical for vacuolar targeting of BbCrpa. The amino acid residue, K (lysine) was identified as a putative mono-ubiquitination site (*Baxter et al., 2005*; *MacGurn et al., 2012*). In order to probe whether K264 residue was required for vacuolar targeting, a site directed mutant, F$^{258}$ASFLPAFLFE$^{268}$ (K264A), was generated. Microscopic visualization revealed that K264 function as a critical residue for proper targeting of B$^{N258-268\,(K264A)}$::mRFP (*Figure 4F*). Western blot analysis further showed that K264 is responsible for the ubiquitination of B$^{N258-268\,(K264A)}$::mRFP (*Figure 4—figure supplement 1G*).

A C-terminal deletion mutant, ΔB$^{C1326-1359}$ (deletion of 1326–1359 aa of BbCrpa), did not affect CsA/FK506 resistance (*Figure 4A–C*, *Figure 4—figure supplement 1A and B*). However, removal of one additional amino acid, Y1325 (tyrosine), resulting in the mutated protein, ΔB$^{C1325-1359}$, led to a significant decrease in resistance to CsA/FK506 (*Figure 4A–C*, *Figure 4—figure supplement 1A and B*). For further investigation of this tyrosine function, we conducted site directed mutagenesis of Y1325 to A (alanine). The mutation led to a significant decrease in CsA/FK506 resistance (*Figure 4A–C*, *Figure 4—figure supplement 1A and B*). Sequence analysis indicated the presence of two additional nearby (in the C-terminus) tyrosine residues: Y1341 and Y1350 (*Figure 4A*). Site directed mutants bearing double substitution mutations of (1) Y1325A-Y1341A, and (2) Y1325A-Y1350A, and a triple substitution mutation: (3) Y1325A-Y1341A-Y1350A resulted in significant decreases in CsA/FK506 resistance as compared to the wild-type protein and as compared to the Y1325A single mutant (*Figure 4A–C*, *Figure 4—figure supplement 1A and B*). These results suggest a key role of these tyrosine residues, in particular, Y1325, for the detoxification activity of BbCrpa.

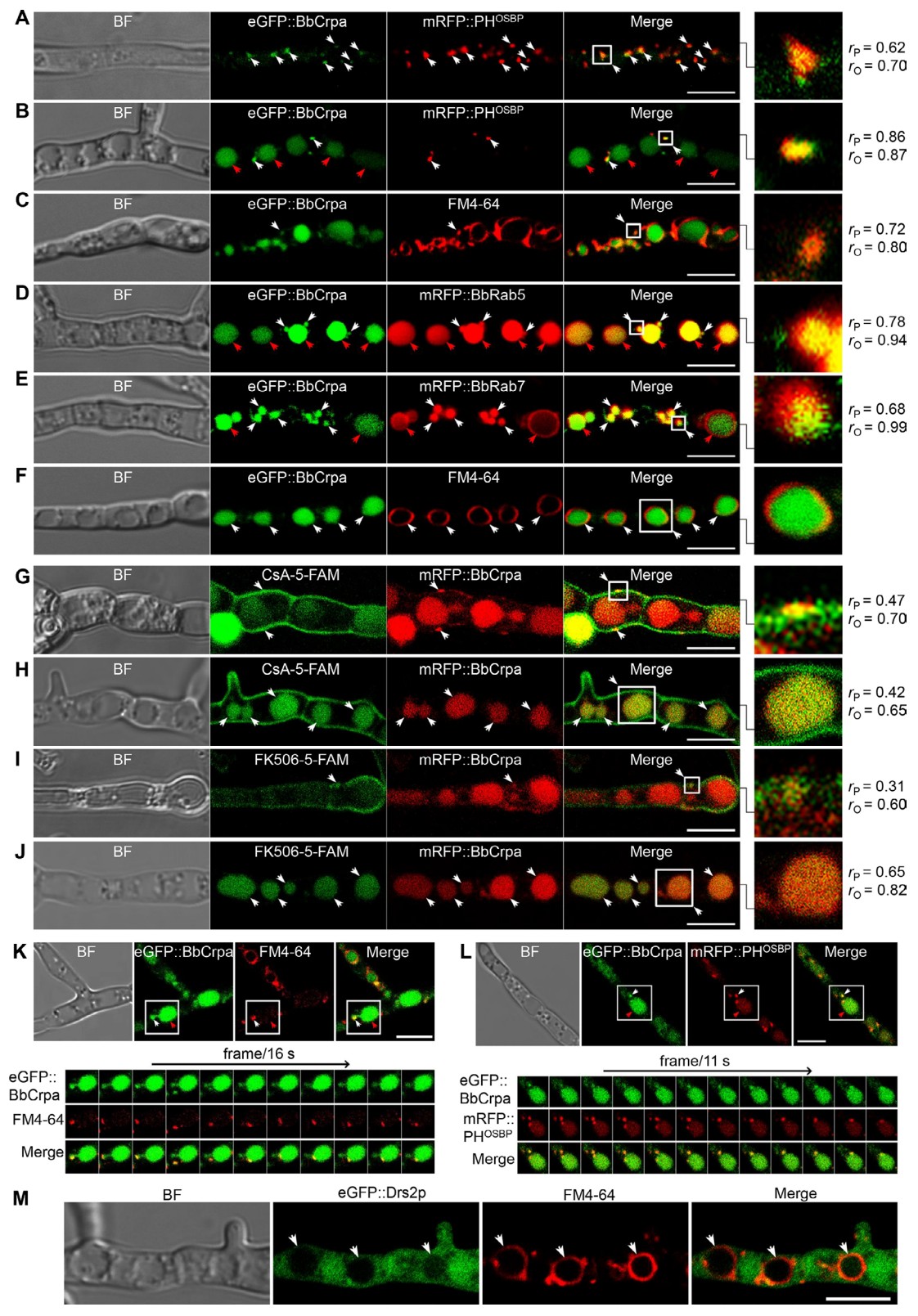

**Figure 3.** Subcellular localization and dynamic trafficking of BbCrpa. (**A and B**) eGFP::BbCrpa colocalizes with mRFP::PH$^{OSBP}$ at TGN (white arrows) and accumulates in vacuoles (red arrows, see also that in (**F**, white arrows)). (**C**) eGFP::BbCrpa accumulates in vesicle (white arrows) that was stained by FM4-64. (**D**) eGFP::BbCrpa colocalizes with mRFP::BbRab5 in (EEs, white arrows) and then accumulates in vacuoles (red arrows, also see in (**F**, white arrows)). (**E**) eGFP::BbCrpa colocalizes with mRFP::BbRab7 in LEs (white arrows) and vacuoles (red arrows). (**F**) eGFP::BbCrpa accumulates in mature vacuoles

*Figure 3 continued*

(arrows) which were stained by FM4-64. (**G**) mRFP::BbCrpa colocalizes with CsA-5-FAM in puncta on the plasma membrane (arrows). (**H**) mRFP::BbCrpa colocalizes with CsA-5-FAM in vacuoles (arrows). (**I**) mRFP::BbCrpa colocalizes with FK506-5-FAM in puncta on the plasma membrane (arrows). (**J**) mRFP::BbCrpa colocalizes with FK506-5-FAM in vacuoles (arrows). (**K**) eGFP::BbCrpa appears in vesicles (white arrows) and moves into vacuoles (red arrows). Time to acquire one image pair was 16 s. (**L**) eGFP::BbCrpa colocalizes with vesicle from TGN which was labeled by mRFP::PH$^{OSBP}$ (white arrows) and transports into vacuole (red arrows). Time to acquire one image pair was 11 s. (**M**) Localization of eGFP::Drs2p in *B. bassiana* cells. eGFP::Drs2p did not accumulate in vacuoles (arrows) which were stained by FM4-64. Scale bars, 5 μm for (**A–M**). The value of Pearson ($r_P$) and Overlap ($r_O$) correlation coefficient shows the extent of colocalization between the two target molecules. The values range between +1 (positive correlation) and –1 (negative correlation).

The online version of this article includes the following video, source data, and figure supplement(s) for figure 3:

**Figure supplement 1.** BbCrpa N-terminally tagged with eGFP maintains its original function and is localized to the apical plasma membrane and Spitzenkörper of *B. bassiana*.

**Figure supplement 1—source data 1.** Growth of target strains at CZP supplemented with CsA normalized to growth at CZP.

**Figure supplement 2.** BFA treatment decreases BbCrpa accumulation in the vacuoles.

**Figure 3—video 1.** Time-lapse imaging of the trajectory of eGFP::BbCrpa (green) and vesicle (red) labeled by FM4-64.
https://elifesciences.org/articles/79179/figures#fig3video1

**Figure 3—video 2.** Time-lapse imaging of the trajectory of eGFP::BbCrpa (green) and vesicle (red) derived from TGN which was labeled by mRFP::PH$^{OSBP}$ (red).
https://elifesciences.org/articles/79179/figures#fig3video2

## Exogenous overexpression of *BbCRPA* in *Arabidopsis thaliana* and *Gossypium hirsutum* increases the resistance to Verticillium wilt disease

The phytopathogenic fungus, *V. dahliae Kleb* is responsible for causing a devastating wilt disease infecting many important trees and crops including elm, cotton, potato, pepper, watermelon, mint, and lettuce (*Subbarao et al., 1996*; *Veronese et al., 2003*). Toxins (*V. dahliae* toxins, VD-toxins) produced by the fungus contribute to wilt symptoms (*Fradin and Thomma, 2006*; *Keen et al., 1972*; *Meyer et al., 1994*). Cinnamyl acetate (CIA) is one of the lipophilic VD-toxins identified (*Laouane et al., 2011*). Detoxification of such mycotoxins might represent an effective way to decrease the damage causing by phytopathogens (*Wang et al., 2020*). Similar to CsA and FK506, CIA is a lipophilic compound. Furthermore, we have shown that the expression of *BbCRPA* in *V. dahliae* was capable of increasing the resistance of the fungus to CsA significantly (*Figure 1D*). The similar feature of CsA, FK506, and CIA in lipophilicity, the cross-species resistance against CsA, FK506 endowed by BbCrpa *in V. dahliae*, as well as the successful application of insecticidal and herbicidal genes from bacteria in genetically modified crops promoted us to test whether BbCrpa could be broadly exploited for detoxification of the fungal mycotoxin CIA in plants. To this end, we generated transgenic *A. thaliana* and *G. hirsutum* in which *BbCRPA* was under control of a constitutive promoter CaMV35S. Transgenic plants were screened by Kalamycin resistance and GUS activity and confirmed by PCR (*Figure 5—figure supplement 1A–D*). Two homologous transgenic *Arabidopsis* lines and two homologous cotton lines with relative high expression level were selected as representative lines for further study. Southern blot results validated *BbCRPA* insertion in chromosomes of *Arabidopsis* and cotton (*Figure 5—figure supplement 1E*). In addition, flanking sequencing of transgenic cotton line *35 S::BbCRPA-B1* indicated that transgene (*35 S::BbCRPA-35S::NTPII::GUS*) was inserted in the A subgenome of 11 chromosome at 120298690. The expression of *BbCRPA* in *A. thaliana* increased tolerance to CIA (*Figure 5A*). With little or no growth was seen in control plants in the presence of 50 μg/ml CIA, under the condition in which plants expressing *BbCRPA* were able to grow (*Figure 5A*). Plant bioassays, infecting either *A. thaliana* or *G. hirsutum* with *V. dahliae* and *Fusarium oxysporum*, revealed a significant reduction in symptom severity (*Figure 5B–G*, *Figure 5—figure supplement 1G–J*). In order to probe the potential mechanism mediating the detoxification of CIA, FITC labeled CIA was fed to plants with sections counterstained with FM4-64 (*Figure 5H,I*). Data showed the promoted accumulation of CIA in vacuoles in *BbCRPA* transformed plant cells, and not in the wild-type parent.

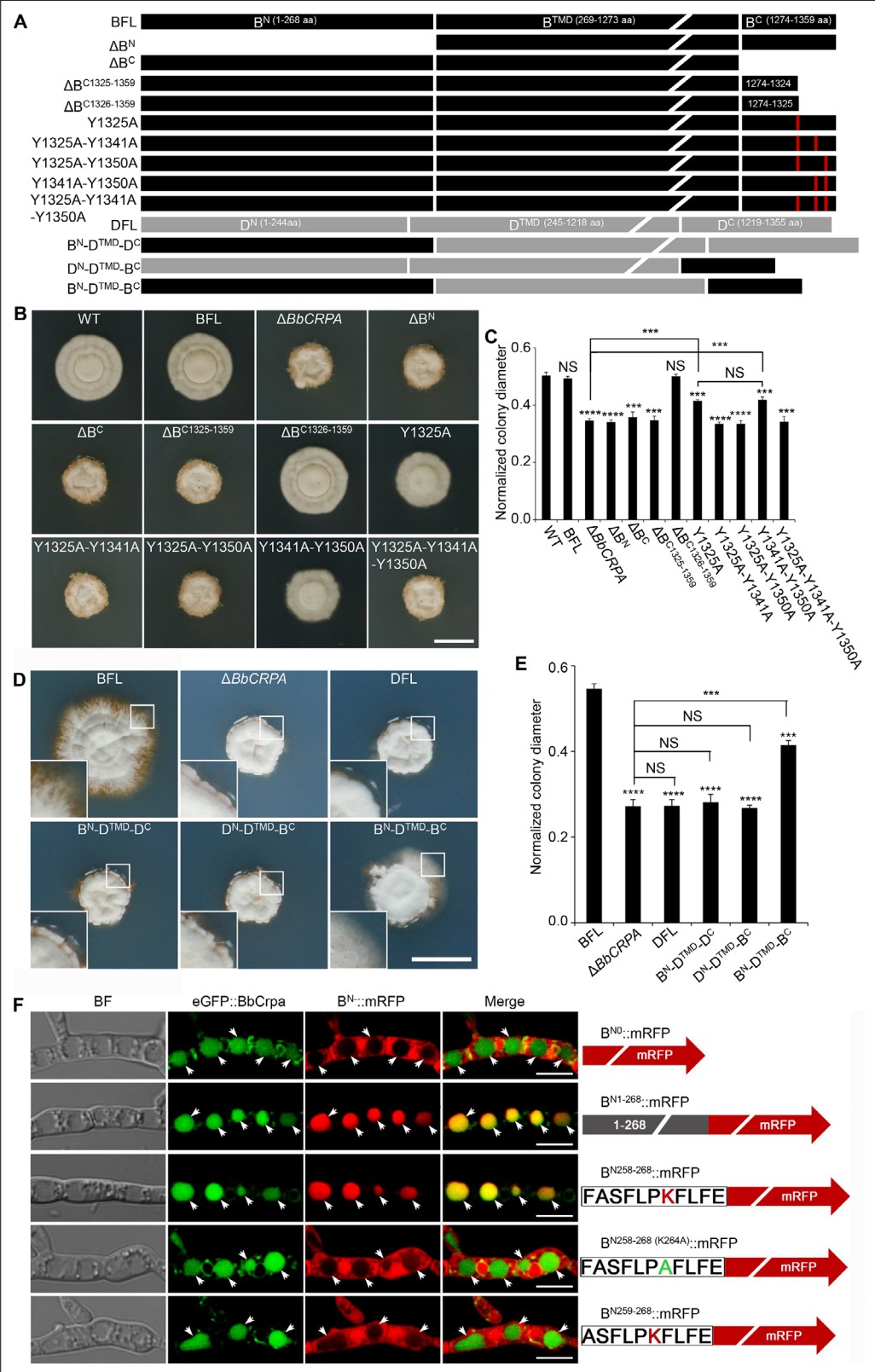

**Figure 4.** Y1325 (Tyr) in C-terminus is critical for detoxification, and the N-terminus is essential for vacuolar targeting. (**A**) Schematic model of N- and C-terminal deletion of BbCrpa and the graft of the N- and C-terminus of BbCrpa with Drs2p. BFL, BbCrpa full length; B^N, BbCrpa N-terminus; B^TMD, BbCrpa transmembrane-domain; B^C, BbCrpa C-terminus; DFL, Drs2p full length; D^N, Drs2p N-terminus; D^TMD, Drs2p transmembrane-domain; D^C, Drs2p

*Figure 4 continued on next page*

*Figure 4 continued*

C-terminus. (**B and C**) BbCrpa C-terminus Y1325 is critical for the detoxification of CsA. All strains were incubated in CZP +CsA (20 μg/ml). (**D and E**) BbCrpa N- and C-terminus are crucial for Drs2p detoxification activity. All strains were incubated in CZA +CsA (20 μg/ml). (**F**) The last 11 amino acid residues of BbCrpa N-terminus contains vacuolar localization signal. BbCrpa N-terminus (B$^{N1-268}$, B$^{N258-268}$) fused with mRFP are obviously colocalized with BbCrpa in vacuoles. When the K264 (Lys) was replaced by Ala (**A**), the guiding function was disappeared. B$^{N-}$::mRFP, mRFP fused with different length of N-terminus of BbCrpa. For CsA sensitivity analysis, plates were spot inoculated with 3 μl conidial suspensions (1×10$^7$ conidia/ml) and incubated at 26°C for about 10 (CZP/CZP +CsA) or 14 (CZA/CZA +CsA) days. The variation in growth rates was shown as [colony diameter CZP/CZA supplemented with CsA]/[colony diameter CZP/CZA]. All experiments were performed in triplicate (n = 3). Data are represented as the mean ± SD. \*\*\*p<0.001; \*\*\*\*p<0.0001 from Student's *t*-test. NS, not significant. Scale bars, 1 cm for (**B and D**) and 5 μm for (**F**).

The online version of this article includes the following source data and figure supplement(s) for figure 4:

**Source data 1.** Growth of target strains at CZP/CZA supplemented with CsA normalized to growth at CZP/CZA.

**Figure supplement 1.** Y1325 (Tyr) in C-terminus is critical for detoxification of FK506, and the N-terminus is essential for vacuolar targeting.

**Figure supplement 1—source data 1.** Growth of target strains at CZP/CZA supplemented with FK506 normalized to growth at CZP/CZA.

**Figure supplement 1—source data 2.** Uncropped western blot.

## Discussion

Mounting reports have documented that P4-ATPases display important roles in vesicle biogenesis, membrane trafficking or remodeling, signal transduction, biotic-/abiotic-stress response, and polarized growth. Nevertheless, little is known about their function in cell detoxification. In the present study, we identify a P4-ATPase gene, *BbCRPA*, from insect disease fungus *Beauveria bassiana* that displays resistance to CsA and FK506. The colocalization of CsA/FK506 and BbCrpa to the puncta on plasma membrane (*Figure 3G,I*) suggests that the toxins are wrapped into vesicles there. Then, the toxins are delivered into vacuoles for compartmentation, which confers the resistance of CsA and FK506 to the fungus.

In eukaryotic cells, another catabolic pathway that sequesters undesired materials is autophagy (*Kaufmann et al., 2014*). In autophagy pathway, the membrane of autophagosome is newly formed (*Barz et al., 2020*). In P4-ATPases-mediated vesicle formation, however, the membrane of the vesicle is from where the P4-ATPase is located on. The P4-ATPases catalyze the translocation of phospholipids from the exoplasmic to the cytosolic membrane leaflet to establish phospholipid asymmetry in biological membranes, and thus to promote budding of transport vesicles. Knocked out *ATG1*, a crucial factor for regulating of autophagosome-vacuole fusion, we found the disruption does not affect the resistance to CsA, as well as the formation of autophagosome in *B. bassiana* (*Supplementary file 1A-C*), suggesting that BbCrpa-mediated detoxification is independent of autophagy pathway.

*B. bassiana* can be dormant in soil for years. As a pathogen, the fungus can infect insects; as an endophyte, it can reside in plants, it can reside in plants (*Ownley et al., 2008*; *Xiao et al., 2012*). To survive, *B. bassiana* evolves mechanisms to protect it from harmful materials produced by microorganisms, insects, or plants. CsA is a secondary metabolite produced by *Beauveria nivea,* another entomopathogenic fungus of *Beauveria* (*Margaritis and Chahal, 1989*). This small lipophilic polypeptide can freely cross the plasma membrane (*Hunt and Morshead, 2010*). The lipophilic macrolide FK506 is produced by the soil borne streptomycete, *Streptomyces tsukubaensis* (*Barreiro et al., 2012*). Both CsA and FK506 bind to their cognate immunophilins, Cyps and FKBPs, to form binary complexes which then block the phosphatase activity of calcineurin, or inhibit the peptidyl-prolyl *cis-trans* isomerase (PPIase) activity of Cyps and FKBPs and thus impair Cyps-/FKBPs-mediated protein folding (*Kang et al., 2008*; *Wang and Heitman, 2005*). Interestingly, although *B. bassiana* and *B. nivea* belong to same genus, no cyclosporin synthetase genes was found in *B. bassiana* genome (https://fungismash.secondarymetabolites.org/#!/start), while no BbCrpa homologue was detected in *B. nivea* (https://blast.ncbi.nlm.nih.gov/Blast.cgi). Therefore, it is acceptable that for competition and survival, *B. bassiana* has developed a unique vesicle-mediated detoxification mechanism against CsA; while *B. nivea,* the toxin producer, has its own orchestrated delivery and timing system, as well as subcellular containment system, to avoid self-harm from the toxin (*Keller, 2015*).

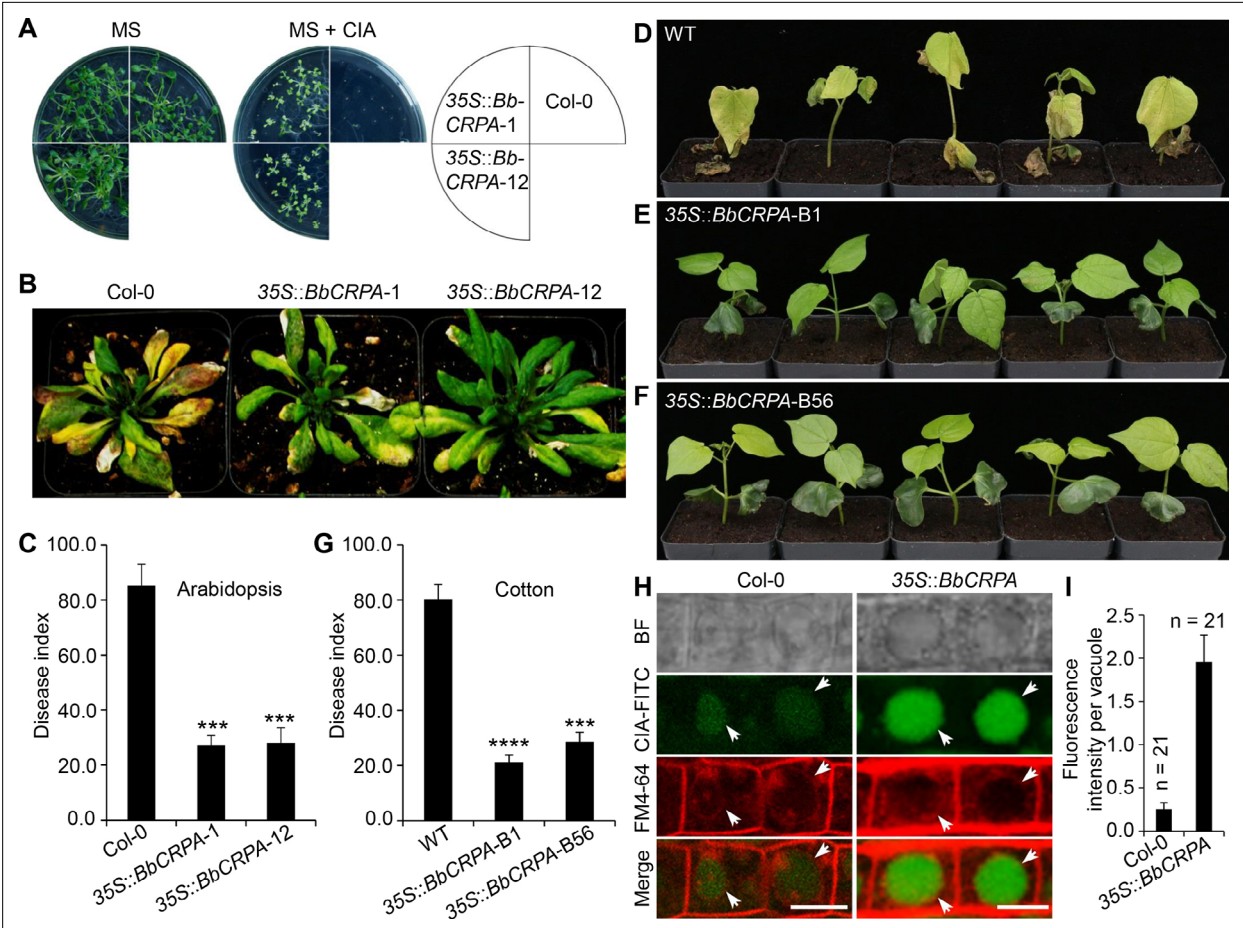

**Figure 5.** Exogenous overexpression of *BbCRPA* in *Arabidopsis* and cotton increases the resistance to *V. dahliae*. (**A**) Exogenous overexpression of *BbCRPA* increases the resistance of *Arabidopsis* to CIA. The wild-type (Columbia, Col-0) and *35 S::BbCRPA* seedlings (T₃) were grown for 14 days on MS medium containing either 0.05% DMSO (control) or 50 μg/ml CIA. (**B and C**) Exogenous overexpression of *BbCRPA* increases the resistance of *Arabidopsis* to *V. dahliae*. Each *Arabidopsis* plant (4–5 leaves) was inoculated with 3 ml *V. dahliae* spore suspension (2×10⁸ spores/ml), and the disease index of transgenic plants and the wild-type plants to *V. dahliae* was evaluated 21 days after inoculation. *35 S::BbCRPA*-1 and *35 S::BbCRPA*-12: two homologous representative transgenic *Arabidopsis* lines.(**D–G**) Exogenous overexpression of *BbCRPA* in cotton (T₂) increases the resistance to *V. dahliae* V991. Ten-day-old cotton seedling roots were infected by *V. dahliae* spore suspension (1×10⁷ spores/ml), and the disease index of the transgenic cottons to *V. dahliae* was evaluated 14 days after inoculation.*35 S::BbCRPA*-B1 and *35 S::BbCRPA*-B56: two homologous representative transgenic cotton lines. (**H and I**) Exogenous overexpression of *BbCRPA* in *Arabidopsis* promotes the accumulation of cinnamyl acetate (CIA) in vacuoles of root cells. Strong fluorescein-labeled CIA (CIA-FITC) signal was observed in vacuoles of transgenic *Arabidopsis* cells (*35 S::BbCRPA*), while only very weak signal was observed in the vacuoles of wild-type *Arabidopsis* cells (arrows) (**H**). Vacuoles (arrows) were indicated by FM4-64 and plants were treated with 5 μg/ml CIA-FITC. The comparison of fluorescent intensity in vacuoles between the wild-type and *35 S::BbCRPA* transgenic *Arabidopsis* was measured by ImageJ (**I**).Data are represented as the mean ± SD. Scale bars, 10 μm. The resistance of plants to *V. dahliae* was estimated by disease index. Data are represented as the mean ± SD for three independent experiments with at least 11 plants per replication. ***p<0.001; ****p<0.0001 from Student's *t*-test.

The online version of this article includes the following source data and figure supplement(s) for figure 5:

**Source data 1.** Verticillium disease index and fluorescent intensity of CIA-FITC in vacuoles.

**Figure supplement 1.** Validation of transgenic *BbCRPA* plants.

**Figure supplement 1—source data 1.** Transcriptional level detection of *BbCRPA* in transgenic plants.

**Figure supplement 1—source data 2.** Validation of transgenic *Arabidopsis* and cotton.

**Figure supplement 1—source data 3.** Disease index analysis.

Although the closest homologue among five yeast P4-ATPases to BbCrpa is Drs2p, functions of BbCrpa are quite different from those of Drs2p. Strong BbCrpa signal was observed in vacuoles, but no Drs2p signal was detected in the vacuoles (*Figure 3F and M*). It has been known that Drs2p is required for AP-1/Clathrin-coated vesicle formation (*Liu et al., 2008*). However, BbCrpa-mediated

resistance to CsA is independent with AP-1 and Clathrin (*Supplementary file 1D and E*). NPFXD motifs can interact with the Sla1p homology domain 1 (SHD1) of Sla1p that serves as the targeting signal recognition factor for $NPFX_{(1,2)}D$-mediated endocytosis (*Howard et al., 2002*; *Liu et al., 2007*). Drs2p has two NPFXD motifs in its C-terminal tail. In contrast, no NPFXD motif is found in the C-terminus of BbCrpa, suggesting that this P4-ATPase may be not involved in NPFXD/Sla1p endocytosis pathway. In addition, our N-terminal serial deletion and mRFP tagging results reveal that the last eleven-amino acid residues ($F^{258}ASFLPKFLFE^{268}$) are capable of guiding protein into the vacuole (*Figure 4F*). When the Lys residue was replaced by Ala, the residues loses the vacuole-target function, implying that the process is associated with ubiquitination modification (*Figure 4F*, *Figure 4—figure supplement 1G*). For Drs2p, no FASFLPKFLFE homologous sequence is present in its N-terminal tail, supporting the previous studies that Drs2p-mediated vesicle transport pathway is between TGN and EE or plasma membrane and TGN, but not including the journey to the vacuole (*Liu et al., 2007*). In N-terminal tail of Drs2p, there are two potential PEST motifs and one poor PEST motif (*Liu et al., 2007*). PEST motif is considered as a signature of proteins that would be degraded through the ubiquitin pathway (*Roth et al., 1998*). However, no typical PEST motif was found in BbCrpa N-terminus. This suggests that Drs2p is the target for ubiquitin-mediated degradation in proteasome while BbCrpa may be involved in the turnover of the membrane protein in TGN membrane or vacuole membrane (*Ciechanover, 1998*; *Rotin et al., 2000*). Most importantly, BbCrpa can confer resistance of CsA/FK506 to the cell while Drs2p cannot (*Figure 4A, D and E*, *Figure 4— figure supplement 1C and D*).

Protein tyrosine phosphorylation/dephosphorylation is an important mechanism for regulating of many cellular processes (*Ghelis, 2011*; *Rosenblum et al., 1995*; *Schlessinger and Ullrich, 1992*). It was reported that CsA/Cyps and FK506/FKBP12 participated in the regulation of protein tyrosine phosphorylation/dephosphorylation (*Bauer et al., 2009*; *Lopez-Ilasaca et al., 1998*). For example, interleukin-2 tyrosine kinase (Itk) is a protein tyrosine kinase of the Tec family. The tyrosine kinase activity of Itk is inhibited by CypA *via* forming a stable complex, and CsA can increase the phosphorylation levels of Itk thus relieving the inhibition of CypA on it (*Brazin et al., 2002*). We show a possible role of Y1325 phosphorylation in detoxification activity of BbCrpa (*Figure 4A–C*). The common target of CsA/Cyp and FK506/FKBP12 is calcineurin, also a PP2B enzyme. Typically, calcineurin is a serine/threonine protein phosphatase. Although it had been reported that calcineurin can also dephosphorylate phosphotyrosine containing proteins (*Carrera et al., 1996*; *Chan et al., 1986*; *Faure et al., 2007*), no PXIXIT nor LXVP motif, the binding motifs of calcineurin (*Shi, 2009*), was found in BbCrpa sequence, suggesting that the phosphorylation/dephosphorylation of Y1325 may not be regulated directly by calcineurin. Does (or how does) CsA/FK506 stimulate the tyrosine phosphorylation of BbCrpa awaits further investigation.

Toxins produced by pathogens are recognized as a pathogenic factor in many plant diseases (*Tsuge et al., 2013*). For plant disease control, cell detoxification strategy can prevent plants from the toxicity, thus ensuring hosts exhibit their innate immunity against the pathogens (*Wang et al., 2020*). In this study, the ectopic expression of *BbCRPA* in *V. dahliae* significantly increases the resistance of the fungus against CsA, demonstrating that this P4-ATPase can display its detoxification ability in other species. We then exogenously overexpressed the gene in cotton and *Arabidopsis*. The transgenic plants exhibit significantly increased resistance to the Verticillium wilt disease (*Figure 5A–G*). Meanwhile, the vacuole-targeted transport of the toxic material (i.e. cinnamyl acetate, CIA) produced by the wilt-disease fungus *V. dahliae* was significantly promoted, indicating that CIA is also the cargo of BbCrpa-mediated vesicle transport in plant cells. Details about what properties the compounds should have and how the P4-ATPases recognize their cargoes await further investigations. Nevertheless, the method described here allows us to identify other P4-ATPases that are able to detoxify various mycotoxins, such as voitoxin, trichothecenes, fumonisins, and ochratoxin A, which are not only involved in the virulence of the phytopathogenic fungi but also extremely harmful for human and animal health.

The P4-ATPase-conferred resistance to CsA/FK506 also suggests a strategy through the vesicle transport associated detoxification to reduce the side effects of the drugs in human organ transplant operation and autoimmune-diseases therapy. Besides, using the fluorescein-labeled compounds and proteins as probes allows us to observe the real time process of cargo delivering as well as dynamic alteration of vesicles, EEs, LEs, and vacuoles in vivo, thus providing a simple and effective platform to

study the subcellular processes of P4-ATPase mediated vesicle transport and the important aspects of the compartmental detoxification in living cells.

In conclusion, our study uncovers the molecular mechanism underlying the inherent resistance of *B. bassiana* against CsA and FK506. Importantly, we show a novel function of P4-ATPases in cellular detoxification. The ability of P4-ATPases to confer cross-species resistance to toxins provides potential opportunities for their utilization in plant disease control and the protection of human health.

## Materials and methods

### Strains and culture conditions

*Beauveria bassiana* wild-type (CGMCC7.34, China General Microbiological Culture Collection Center, CGMCC) and mutant strains were grown on Czapek-Dox broth (CZB) or agar (CZA) (233810, BD-Difco, USA), or CZB or CZA supplemented with 0.25% (wt/vol) tryptone (CZP). *Verticillium dahliae* strain V991 (kindly gifted by Prof. Guiliang Jian) was grown on Potato Dextrose broth (PDB) or agar (PDA) (254920, BD-Difco). *Escherichia coli* DH5α (9057, Takara) and *Agrobacterium tumefaciens* AGL-1 (Lab stock) were used for routine DNA manipulations and fungal transformations, respectively. *A. tumefaciens* GV3101 (Lab stock) and LBA4404 (Lab stock) were used for *Arabidopsis* (Col-0, CS60000, *Arabidopsis* Biological Resource) and cotton (*cv.* Jimian 14, kindly donated by Prof. Zhiying Ma) transformation, respectively.

### Inhibition ring assay

Filamentous fungi used in inhibition ring assay include *B. bassiana*, *Metarhizium anisopliae* (kindly gifted by Prof. Weiguo Fang), *Botrytis cinerea* (3.4584, CGMCC), *Alternaria brassicae* (3.7804, CGMCC), *Alternaria brassicicola* (3.7805, CGMCC), *Aspergillus nidulans* (3.15737, CGMCC), *Alternaria solani* (Lab stock). The fungi were inoculated into PDA medium except *B. cinerea* (regeneration medium: 0.1% yeast extract, 24% sucrose) and *A. nidulans* (modified PDA medium containing 0.11% uracil and 0.12% uridine) at 26°C for about 10–14 days for conidia harvested. Then, for each strain, 300 μl of fungal conidial suspensions ($1 \times 10^7$ conidia/ml in 0.05% Tween-80, A600562, Sangon Biotech, China) was added into 60 ml PDA (approximately 45°C, *A. nidulans* conidial suspension was added into the modified PDA medium) medium, mixed evenly and poured into three Petri Dishes (diameter = 90 mm), averagely. After the media completely solidified, punch four holes of 5 mm diameter equally distant apart in Petri Dishe. Each well was added with 5 μl different concentrations of CsA (B1922, APEXBIO) or FK506 (B2143, APEXBIO), and the dishes were incubated at 26°C for about 4–7 days.

### Screening of CsA-Sensitive mutants and isolation of the target gene

*B. bassiana* random insertion (T-DNA) library was constructed as described previously (*Fang et al., 2004*). Plasmid pK2 was used as frame vector. *Bar::Gus* fusion reporter gene was placed into pK2 between the *A. nidulans gpdA* promoter and *trpC* terminator. CZP medium containing CsA (20 μg/ml) was used to screen CsA-sensitive mutants. The desired genomic sequence was isolated by PCR walking using the YADE method as described previously (*Fang et al., 2005*).

### Construction of Δ*BbCRPA* and complementation strains

A list of primers used in the nucleic acid manipulations was given in *Supplementary file 2*. LA Taq (RR002A, TaKaRa) or PrimeSTAR MAX Premix (R045, TaKaRa) was used for the generation of PCR products. The resultant products were cloned into target vectors using T4 DNA ligase (CV0701, Aidlab).

The construct of *BbCRPA* gene disruption was generated by homologous recombination. Plasmid pK2-*Bar* containing herbicide (phosphinothricin, P679, PhytoTech) resistance gene (*Bar*) that was sandwiched between *A. nidulans trpC* promoter and *trpC* terminator was used as backbone to construct the transformation vector. The vector was constructed as follows: a 5′ end 1.294 kb sequence (1053–2346 bp) and a 3′ end 1.214 kb (2897–4110 bp) of *BbCRPA* were amplified by PCR (LA Taq) using *B. bassiana* genomic DNA as template with primer pairs *BbCRPA*LB-F/*BbCRPA*LB-R and *BbCRPA*RB-F/*BbCRPA*RB-R, respectively. The 3′ end PCR product flanking with *Xba*I and *Hind*III restriction sites was digested with *Xba*I (FD0684, Thermo Fisher Scientific) and *Hind*III (FD0504, Thermo Fisher Scientific), and then linked with *Bar* cassette to form *Bar-BbCRPA*3′. Similarly, *BbCRPA* 5′ end PCR product that contains *Eco*RI (FD0274, Thermo Fisher Scientific) sites was fused with *Bar-BbCRPA*3′ to generate

pK2-*BbCRPA*5'-*Bar*-*BbCRPA*3'. The resulting vector was transformed into *A. tumefaciens* AGL-1, which was then used to transform wild-type *B. bassiana* as described previously (*Fang et al., 2004*). Mutant colonies were single spore isolated and the correct integration event was verified by PCR using 2×Taq Master Mix (E005-2b, Novoprotein) with primers *MCS*-F/*MCS*-R and qRT-PCR with primer pair *BbCRPA*ex-F/*BbCRPA*ex-F.

The selection marker gene, *Sur*, conferring sulfonylurea (chlorimuron-ethyl, J66605, Alfa Aesar, USA) resistance was used for complementation vector construction. The gene was amplified from pCB1536 with primer pair *Sur*-F/*Sur*-R (*Zhang et al., 2010*). The PCR product was digested with *Not*I (FD0594, Thermo Fisher Scientific) and *Bam*HI (FD0055, Thermo Fisher Scientific), and then inserted into modified PUC-T vector (D2006, Beyotime) between *A. nidulans trpC* promoter and *A. nidulans trpC* terminator. The vector was digested with *Eco*RI and *Hin*dIII and inserted into the corresponding sites of pK2-*Bar*, replacing the *Bar* gene, to generate pK2-*Sur*. The complementation vector, pK2-*Sur*-*BbCRPA*, was constructed using a 5998 bp fragment that contains the entire ORF (4153 bp including a 73 bp intron), 1072 bp of upstream, and 773 bp of downstream sequences. The fragment was amplified *via* PCR (PrimeSTAR MAX Premix) with primers *Com*-F and *Com*-R using *B. bassiana* genomic DNA as template. The PCR product was digested with *Spe*I (FD1254, Thermo Fisher Scientific) and *Xba*I and then inserted into pK2-*Sur* to form pK2-*Sur*-*BbCRPA*. The resulting vector was used for Δ*BbCRPA* transformation.

## Site-directed mutagenesis of BbCrpa and N/C-terminal tail exchange between BbCrpa and Drs2p

For the change of Ile (562) to Glu (I562E), and Asp (614) to Arg (D614R) of BbCrpa, the fragments (1 and 2nd of I562E and D614R) were cloned using paired primers *BbCRPA*-F/*Bb* (*I562E*)-first-R (I562E-1st), *Bb* (*I562E*)-second-F/*BbCRPAN*-R (I562E-2nd), *BbCRPA*-F/*Bb* (*D614R*)-first-R (D614R-1st), *Bb* (*D614R*)-second-F/*BbCRPAN*-R (D614R-2nd) *via* PCR (PrimeSTAR MAX Premix). The full-length fragments (1st + 2nd) of I562E and D614R were integrated by overlap extension PCR. For BbCrpa N/C-terminus deletion and C-terminus site-directed mutagenesis, the target fragments were cloned with primer pairs *Bb* (Δ*N*)-F/*BbCRPAN*-R (ΔB[N]), *BbCRPA*-F/*Bb* (Δ*C*)-R (ΔB[C]), *BbCRPA*-F/*Bb* (Δ*C1325-1359*)-R (ΔB[C1325-1359]), *BbCRPA*-F/*Bb* (Δ*C1326-1359*)-R (ΔB[C1326-1359]), *BbCRPA*-F/*Bb* (*1325*)-R (Y1325A), *BbCRPA*-F/*Bb* (*1325–1341*) R (Y1325A-Y1341A), *BbCRPA*-F/*Bb* (*1325–1350*) R (Y1325A-Y1350A), *BbCRPA*-F/*Bb* (*1341–1350*) R (Y1341A-Y1350A), *BbCRPA*-F/*Bb* (*1325-1341-1350*)-R (Y1325A-Y1341A-Y1350A). All the PCR products were digested with *Not*I and *Bam*HI and then inserted into the modified PUC-T vector, making it sandwiched by *B. bassiana gpdB[1153]* promoter and *A. nidulans trpC* terminator, respectively. Then, all vectors were digested with *Xba*I and *Spe*I, and then inserted into pK2-*Sur*. The resulting vectors were transformed into *A. tumefaciens* AGL-1 and subsequently used to transform Δ*BbCRPA*.

For the N/C-terminal tail exchange between BbCrpa and Drs2p, BbCrpa full length sequence (BFL), BbCrpa N-terminus (B[N]), transmembrane domains (B[TMD]), C-terminus (B[C]) were cloned using paired primers *BbCRPAN*-F/*BbCRPAN*-R, *BbCRPA*-F/*BbN*-R, *BbTMD*-F/*BbTMD*-R, *BbC*-F/*BbCRPA*-R, respectively. Drs2p full length sequence (DFL), Drs2p N-terminus (D[N]), transmembrane domains (D[TMD]), C-terminus (D[C]) were cloned from *Saccharomyces cerevisiae* cDNA using paired primers *DRS2*-F/*DRS2*-R, *DRS2*-F/*DRS2N*-R, *DRS2TMD*-F/*DRS2TMD*-R, *DRS2C* -F/*DRS2*-R, respectively. Overlap extension PCR was performed for the assemblage of different fragments (B[N]-D[TMD]-D[C], D[N]-D[TMD]-B[C], B[N]-D[TMD]-B[C]). All the PCR products were digested by *Not*I-*Bam*HI or *Not*I, and then inserted into modified pUC-T vector, making it sandwiched with *B. bassiana gpdB[1153]* promoter, and *A. nidulans trpC* terminator, respectively. All the integrate expression elements were cloned with primers *P2*-F/*T2*-R and the products were digested with *Spe*I and inserted into pK2-*Sur*. The resulting vectors were used for Δ*BbCRPA* transformation.

## Construction of eGFP/mRFP fusion proteins

For N-terminal tagging with eGFP, the coding sequence of BbCrpa was cloned from *B. bassiana* cDNA using primer pair *BbCRPAN*-F/*BbCRPAN*-R and digested with *Not*I and *Bam*HI, and then inserted into the modified PUC-T vector to form PUC-*BbCRPA*. The coding sequence of enhanced green fluorescent protein (eGFP) was amplified from plasmid eGFP-C1 (6084–1, Clontech) using primers *eGFPN*-F/*eGFPN*-R. The resultant fragment was treated with *Not*I, and then cloned into the accomplished

vector PUC-*BbCRPA* to generate PUC-*eGFP::BbCRPA*, in which the expression of *eGFP::BbCRPA* is driven by *B. bassiana* $gpdB^{1153}$ promoter and stopped by *A. nidulans trpC* terminator. Treated the vector with *Xba*I and *Spe*I, and the resultant fragment was inserted into pK2-*Sur* and pK2-*Bar*, respectively. The resulting vectors were transformed into Δ*BbCRPA* and wild-type *B. bassiana*, respectively. For N-terminal tagging with mRFP (monomeric red fluorescent protein), the coding sequences of mRFP was cloned from plasmid p1793 (*Pantazopoulou and Peñalva, 2009*) with primer pair *mRFP*N-F/*mRFP*N-R and *BbCRPA* was cloned using *B. bassiana* cDNA as template with primer pair *BbCRPA* (*mF*)-F/*BbCRPA*N-R. The PCR products were fused by overlap extension PCR to link *mFRP* and *BbCRPA* together. The fusing fragment was digested with *Not*I and *Bam*HI, and then inserted into the modified PUC-T to form PUC-*mRFP::BbCRPA*, the expression of which is driven by *B. bassiana* $gpdB^{1153}$ promoter and sotpped by *A. nidulans trpC* terminator. Then, the vector was digested with *Xba*I and *Spe*I, and inserted into pK2-*Sur*. The resulting vector was used for the transformation of Δ*BbCRPA*.

For labeling late Golgi, $gpdA^{mini}::mRFP::PH^{OSBP}$ was amplified from plasmid p1793 using primers *mPH*-F and *mPH*-R. The product was digested with *Xba*I, and then inserted into pK2-*Sur*. The resulting vector was transformed into the wild-type and *eGFP::BbCRPA* strains, respectively.

For labeling EE (early endosome) and LE (late endosome), *B. bassiana* endogenous small GTPases BbRab5 and BbRab7 were cloned using the primer pairs *BbRab5*-F/*BbRab5*-R and *BbRab7*-F/*BbRab7*-R, respectively. The coding sequence of mRFP was amplified from plasmid p1793 using primers *mRFP*N-F/*mRFP*N-R. The fragment of mRFP was fused with BbRab5/BbRab7 by overlap extension PCR. The resulting fusion sequences were cloned into modified PUC-T vector to form PUC-*mRFP::BbRab5* and PUC-*mRFP::BbRab7*, respectively. PUC-*mRFP::BbRab5* was digested with *Xba*I and *Spe*I, and then inserted into pK2-*Sur*. The fragment of $gpdB^{1153}::mRFP::BbRab7::trpC$ was cloned from PUC-*mRFP::BbRab7* using primers *P1*-F/*T1*-R. The resulting PCR product was digested with *Xba*I, and then inserted into pK2-*Sur*. All the resulting vectors were transformed into *eGFP::BbCRPA* strain and wild-type *B. bassiana*.

For Drs2p localization observation, the coding sequences of eGFP and Drs2p were amplified from plasmid eGFP-C1 and *Saccharomyces cerevisiae* cDNA using primer pairs *eGFP*N-F/*eGFP* (*NF*)-R and *DRS2* (Fu)-F/*DRS2*-R, respectively. eGFP and Drs2p were fused by overlap extension PCR. The resulting product, *eGFP::DRS2*, was digested with *Not*I, and then inserted into the modified PUC-T. The fragment of $gpdB^{1153}::eGFP::DRS2::trpC$ was cloned from PUC-*eGFP::DRS2* using primers *P2*-F/*T2*-R. The resulting product was digested with *Spe*I, and then inserted into pK2-*Sur*. The resulting vector was transformed into Δ*BbCRPA*.

For BbCrpa full-length N-terminus ($B^{N1-268}$) tagging with mRFP, the coding sequences of $B^{N1-268}$ and mRFP were generated by PCR with primer pairs *BbCRPA*N-F/*N268*-R, *mRFP*-F/*mRFP*-R, respectively. The products were fused by overlap extension PCR. The fusion sequence was then digested with *Not*I and *Bam*HI, and inserted into the modified PUC-T to form PUC-$B^{N1-268}$::mRFP between *B. bassiana* $gpdB^{1153}$ promoter and *A. nidulans trpC* terminator. For tagging $B^{N0}$, $B^{N258-268}$, $B^{N258-268\ (K264A)}$, and $B^{N259-268}$ with mRFP, the coding sequences of them were cloned by PCR with primer pairs *mRFP*N-F/*mRFP*-R, *N258-268*-F/*mRFP*-R, *N258-268* (K-A)-F/*mRFP*-R, *N259-268*-F/*mRFP*-R using plasmid PUC-$B^{N1-268}$::mRFP as template. All the PCR products were digested with *Not*I and *Bam*HI, and then inserted into the modified PUC-T. These modified PUC-T vectors were digested with *Xba*I and *Spe*I, and then inserted into pK2-*Sur*. Finally, the resulting vectors were transformed into eGFP::BbCrpa strain.

## Ectopic expression of *BbCRPA* in *V. dahliae*

For *V. dahliae* transformation, BbCrpa coding sequence was cloned with primer pair *BbCRPA*N-F/*BbCRPA*N-R and digested with *Not*I and *Bam*HI. The resultant fragment was inserted into the modified PUC-T vector. Then, the vector was digested with *Xba*I and *Spe*I. The resulting fragment was inserted into pK2-*Hyg*, a vector with hygromycin B (10687010, Thermo Fisher Scientific)-resistance-encoding gene as a selection marker. The resulting vector was transformed into *A. tumefaciens* AGL-1 and subsequently used to transform the wild-type *V. dahliae* as described previously (*Zhou et al., 2013*).

## Ectopic expression of *BbCRPA* in *Arabidopsis* and cotton

For *Arabidopsis* and cotton transformation, BbCrpa coding sequence was cloned by PCR with primer pair *pBbCRPA*-F/*pBbCRPA*-R. The product was digested with *Bam*HI/*Spe*I, and inserted into modified pCAMBIA2300 (VT1383, YouBio) (PLGN) between cauliflower mosaic virus *35* S promoter and *A. tumefaciens Nos* terminator. The resulting vector was transformed into *A. tumefaciens* GV3101 and LBA4404, and then tansferred into *Arabidopsis* and cotton. *Arabidopsis* transformation was carried out using *Agrobacterium tumefaciens* strain GV3101 by floral-dip method (*Clough and Bent, 2010*; *Harrison et al., 2006*). Homozygous transgenic plants were obtained by kanamycin (50 µg/mL) selection and PCR validation in the segregative generation (T$_1$), and propagated by selfing. Non-transgenic plants were used as null control. Cotton transformation was carried out using *A. tumefaciens* strain LBA4404 by the method as described by *Luo et al., 2007*. Transformates were screened by kanamycin resistance and verified by GUS activity. BbCrpa transcriptional level was examined by quantitative RT-PCR. In T$_1$ generation, homologous transgenic BbCrpa cotton lines were determined by GUS staining and PCR. The non-transgenic plants segregated from selfed progenies were used as control. Homozygous transgenic plants were used for further experiments.

## Phenotypic assays

Fungi were grown on CZP (agar) (*B. bassiana*) or CZA (*B. bassiana*) or PDA (*V. dahliae*) supplemented with CsA or FK506 typically by inoculation of 3 µl of conidial suspensions ($1 \times 10^7$ conidia/ml in 0.05% tween-80) onto the center of agar plates. Plates were incubated at 26°C for about 10–14 days. For growth inhibition rates assays of different strains, the data were shown as (colony diameter CZP/CZA supplemented with CsA/FK506)/(colony diameter CZP/CZA).

## Gene expression analysis

For *BbCRPA* expression analysis, 1 µg total RNA was reverse-transcribed using PrimeScript RT reagent Kit (RR047A, TaKaRa). Quantitative RT-PCR (qRT-PCR) was performed using a CFX96 Real-Time System (Bio-Rad). PCR reactions were performed in 96-well plates as follows: 10 µl iQSYBR Green Supermix (1708882, Bio-Rad), 500 nM forward and reverse primers, and 1 µl 1:7 diluted cDNA template. All experiments were performed thrice. *γ-actin* (for *B. bassiana*), *AtActin*2 (for *Arabidopsis*) and *GhHis*3 (for cotton) were used as the internal reference. The regular PCR cycling conditions were as follows: 3 min at 95 °C, followed by 40 cycles of 10 s at 95°C, 30 s at 56°C and 30 s at 72°C. In order to verify the specificity of the primers, a melt curve analysis was performed for quality assurance. Relative expression of the target gene was normalized to the quantity of the reference gene (normalized fold expression) and processed in CFX Manager 3.1 software (BioRad). Primers used for qRT-PCR analysis are given in *Supplementary file 2*.

## Fluorescence labeling of CsA, FK506, and CIA

CsA and FK506 were labeled with 5-Carboxyfluorescein (5-FAM, HY-66022, MCE). Cinnamyl acetate (CIA, 166170, Sigma-Aldrich) was labeled with fluorescein isothiocyanate (FITC, HY-66019, MCE). The labeled molecules were produced by Fanbo Biochemicals Co. Ltd. (Haidian HighTech Business Park, Beijing, China).

## Sample preparation for imaging

All experiments for imaging of *B. bassiana* were performed in CZB medium, with external supplements added as needed. All the fluorescent strains were precultured in CZB (200 rpm, 26°C), 48–72 hr aged hyphae were harvested from CZB medium (10, 000 rpm, 5 min, 4°C) and resuspended in PBS (NaCl 8 g/l, KCl 0.2 g/l, Na$_2$HPO$_4$·12 H$_2$O 3.63 g/l, KH$_2$PO$_4$ 0.24 g/l, pH 7.4, typically 0.1 ml). For video acquisition, 48–72 hr aged hyphae were harvested and observed second by second. FM4-64 (F34653, Thermo Fisher Scientific) was used to stain the membrane of vacuole, prevacuolar (PVC)/multivesicular body (MVB) and vesicle, and the protocol was as described previously (*Lewis et al., 2009*) with slight modification. Briefly, samples were resuspended into HBSS (Hank's balanced salt solution, NaCl 8 g/l, KCl 0.4 g/l, KH$_2$PO$_4$ 0.06 g/l, Na$_2$HPO$_4$.12H$_2$O 0.121 g/l, Glucose 1 g/l, pH 7.2) containing a final concentration of 8 µM FM4-64 and then incubated at 4°C for 40–60 min. For FM1-43 (T35356, Thermo Scientific) staining, samples were resuspended into HBSS containing a final concentration of 20 µM and then incubated at 4°C for 60 min. For the observation of 5-FAM labeled CsA and FK506 in

the fungus, the labeled molecules were added to the CZB medium to a final concentration of 7.5 µg /ml (CsA) and 6 µg/ml (FK506), respectively, and incubated with fungal cells for 48–72 hr. Finally, the samples were washed with PBS twice for imaging. For the observation of FITC labeled CIA in the plant, *Arabidopsis* seedlings were incubated in liquid Murashige and Skoog medium (MS, M519, Phytotech) (1.5% [w/v] Sucrose, pH 5.8) containing FITC-CIA (5 µg/ml) and FM4-64 (4 µM) at 22°C for 8 hr and the samples were washed with ddH$_2$O 3–4 times for imaging.

## Image acquisition

For confocal microscopy, an inverted confocal laser scanning microscope (FV1000, Olympus) was used. For the observation of fluorescent signals of eGFP, 5-FAM, and FITC, an argon ion laser (Ex = 488 nm, Em = 515–530 nm) was used. For the observation of mRFP and FM4-64, fluorescent signals were acquired using a He-Ne laser (Ex = 559 nm, Em = 570–670 nm). Finally, all the confocal images were captured with FV10-ASW 3.0 Viewer software (Olympus).

## Insect bioassays

Fourth-instar larvae of *Galleria mellonella* were used as target insects for bioassays. Larvae were immersed into suspensions ($2\times10^7$ conidia/ml in 0.05% tween-80) derived from Δ*BbCRPA* and the wild-type for 15 s. Excess solution was removed by treating with paper towel. All treated larvae were transferred into growth chamber at 26 °C for 15 hr:9 hr (light:dark cycle) with 70% relative humidity. The mortality was recorded every 12 hr. Each treatment was performed in triplicate with 30–40 insects with at least two independent batches on conidia. Kaplan-Meier curves were used for analyzing the survival data and a log rank test was used to analyze the difference between Δ*BbCRPA* and the wild-type.

## Western blot and immunoprecipitation

For the immunoprecipitation (IP) and immunoblot assays, the total protein was extracted according to the extraction kit (BB-3136, BestBio). IP was performed according to the manufacturer's protocol (SA079001, Smart-lifesciences) with silght modification. Briefly, the protein complexes were isolated by binding to the anti-RFP affinity beads 4FF, followed by two washes with balanced solution (50 mM Tris, 0.15 M NaCl, pH 7.4). Finally, 200 µl PBS with 5×SDS PAGE loading buffer was added to the complexes and incubated at 95°C for 10 min and the affinity beads were collected by centrifugation (5000 *g*, 1 min). The supernatants that contain the eluted targets were analyzed by immunoblotting with anti-Ub (ubiquitin, PTM-1107, PTM-BIO) and anti-RFP antibodies (MA5-15257, Thermo Fisher Scientific).

## Southern blot

Southern blot was performed according to DIG High Prime DNA Labelling and Detection Starter Kit II (11585614910, Roche). Briefly, 30 µg DNA from the leaves of wild-type and transgenic plants were digested with *Hind*III and subjected to DNA electrophoresis with 0.8% agarose gel. Probe was prepared from the purified PCR product of the *BbCRPA* gene. The labelling of probe, hybridization and detection were performed according to the manufacturer's instructions. The primer pair used for southern blot is given in *Supplementary file 2*.

## Assays for CIA tolerance and disease resistance

Homozygous transgenic T$_3$ *Arabidopsis* plant (4–5 leaves) was inoculated with pathogen spore suspensions (3 ml for *V. dahliae*, 5 ml for *Fusarium oxysporum* f. sp. *vasinfectum* race 7) spore suspensions ($2\times10^8$ spores/ml for *V. dahliae* V991, $1\times10^8$ spores/ml for *V. dahliae* race L2-1, and $2\times10^8$ spores/ml for *F. oxysporum*) and then all *Arabidopsis* plants were transferred into growth chamber at 22 °C for 16 hr:8 hr (light:dark cycle) with 70% relative humidity. The resistance of the plants to pathogens was evaluated 21 days later after inoculation.

For cotton, homozygous T$_2$ transgenic plants were used for disease resistance assays. For the assay to *V. dahliae*, cotton seedlings (10-day-old for *V. dahliae* V991 infection, 21-day-old for *V. dahliae* race L2-1 infection) were treated with pathogen spore suspensions ($1\times10^7$ spores/ml for *V. dahliae* V991, $1\times10^8$ spores/ml for *V. dahliae* race L2-1) according to the method as described previously (*Fradin et al., 2009*) and then the plants were transferred into growth chamber at 26 °C for 16 hr:8 hr

(light:dark cycle) with 70% relative humidity. The disease index of the cotton plants to *V. dahliae* was evaluated 14 (for *V. dahliae* V991 infection) or 21 (for *V. dahliae* race L2-1 infection) days after inoculation. For disease resistance assays of cotton to *F. oxysporum*, cotton seedlings prepared to inoculate were cultivated in a controlled environment chamber at 26 °C for 16 hr:8 hr (light:dark cycle) with 70% relative humidity. 21-day-old cotton seedling roots were gently uprooted, the root tips were cut and dipped into *F. oxysporum* spore suspension ($2\times10^8$ spores/ml) for 3 hr, and then the plants were transplanted into sterilized soil for observation of the symptom of Fusarium wilt. The symptom of infected plants was evaluated by different grades of disease: 0, health plant; 1, 0–25% chlorotic or necrotic leaves; 2, 25–50% chlorotic or necrotic leaves; 3, 50–75% chlorotic or necrotic leaves; 4, 75–100% chlorotic or necrotic leaves or no leaf left or dead plant. The disease index was calculated according to the following formula: DI = [$\sum$(disease grades ×number of infected plants)/(total checked plants ×4)]×100 (*Zhang et al., 2012*). For CIA resistance assays of *Arabidopsis*, *Arabidopsis* seeds were incubated on MS plates supplemented with either 0.05% DMSO (control) or CIA (50 µg/ml) for 14 days.

## Statistical analyses

Statistical analyses were performed with a Student's *t*-test or a log rank test. Significance was defined as **p<0.01; ***p<0.001; ****p<0.0001. Differences with a p value of 0.05 or less were considered significant. For insect survival study, Kaplan-Meier survival curve was generated and analyzed for statistical significance with GraphPad 5.0. Statistical details for each experiment can be found in the Figure Legends.

## Acknowledgements

This work was supported by the Chinese Ministry of Science and Technology of China (2016YFD0100505), National Transgenic New Species Breeding Major Project of China (2016ZX08005-003-004 to YP), and National Major Project of Breeding of China (2018YFD0100403 to XL). We thank Prof. Qixiang Guo (School of Chemistry and Chemical Engineering, Southwest University, Chongqing, China) for his help in CsA/FK506 labeling, and Dr. Zhibing Luo (Biotechnology Research Center, Southwest University, Chongqing, China) for his assistance in the construction of T-DNA insertion mutant pool in *Beauveria bassiana*. We are grateful to Dr. Miguel A Peñalva (Department of Physical and Chemical Biology, Centro de Investigaciones Biológicas, Consejo Superior de Investigaciones Científicas, Ramiro de Maeztu 9, 28040 Madrid, Spain) for providing plasmid p1793, Prof. Weiguo Fang (Institute of Microbiology, Zhejiang University, Hangzhou, China) for the gift of *Metarhizium anisopliae* strain, Guiliang Jian (Institute of Plant Protection, Chinese Academy of Agricultural Sciences, Beijing, China) for the gift of *Verticillium dahliae* strain V991, Prof. Zhiying Ma (State Key Laboratory of North China Crop Improvement and Regulation, North China Key Laboratory for Crop Germplasm Resources of Education Ministry, Hebei Agricultural University, Hebei, China) for the gift of *Gossypium hirsutum*, *cv.* Jimian 14 and *V dahliae* race L2-1, and Prof. Longfu Zhu (National Key Laboratory of Crop Genetic Improvement, Huazhong Agricultural University, Wuhan, Hubei, China) for the gift of *Fusarium oxysporum* f. sp. *vasinfectum* race 7.

## Additional information

### Funding

| Funder | Grant reference number | Author |
| --- | --- | --- |
| Chinese Ministry of Science and Technology of China | 2016YFD0100505 | Yan Pei |
| National Transgenic New Species Breeding Major Project of China | 2016ZX08005-003-004 | Yan Pei |
| National Major Project of Breeding of China | 2018YFD0100403 | Xianbi Li |

| Funder | Grant reference number | Author |
|--------|------------------------|--------|

The funders had no role in study design, data collection and interpretation, or the decision to submit the work for publication.

## Author contributions

Yujie Li, Hui Ren, Conceptualization, Data curation, Software, Formal analysis, Validation, Investigation, Visualization, Writing – original draft, Writing – review and editing; Fanlong Wang, Resources, Data curation, Software, Formal analysis, Investigation, Visualization, Methodology; Jianjun Chen, Lian Ma, Yang Chen, Data curation, Software, Formal analysis, Validation, Methodology; Xianbi Li, Data curation, Software, Funding acquisition, Validation, Investigation; Yanhua Fan, Dan Jin, Data curation, Software, Validation, Investigation, Methodology; Lei Hou, Yonghong Zhou, Software, Validation, Investigation, Methodology; Nemat O Keyhani, Conceptualization, Resources, Validation, Writing – review and editing; Yan Pei, Conceptualization, Resources, Formal analysis, Supervision, Funding acquisition, Validation, Investigation, Project administration, Writing – review and editing

## Author ORCIDs

Yujie Li  http://orcid.org/0000-0001-9293-6233
Yan Pei  http://orcid.org/0000-0001-8317-6199

## Decision letter and Author response

Decision letter https://doi.org/10.7554/eLife.79179.sa1
Author response https://doi.org/10.7554/eLife.79179.sa2

---

# Additional files

## Supplementary files

• Supplementary file 1. Loss of autophagy-related protein BbAtg1, adaptor protein BbAP-1, or coat protein clathrin does not affect CsA resistance.

• Supplementary file 2. Primers used in this study.

• MDAR checklist

• Source data 1. Growth of target strains at CZP/CZA supplemented with CsA normalized to growth at CZP/CZA. File for the primary data corresponding to *Supplementary file 1B and E*.

## Data availability

All data generated or analysed during this study are included in the manuscript and supporting file.

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

# Appendix 1

**Appendix 1—key resources table**

| Reagent type (species) or resource | Designation | Source or reference | Identifiers | Additional information |
|---|---|---|---|---|
| Strain, strain background (*Agrobacterium tumefaciens*) | AGL-1 | Lab stock | | Fungal transformations, Biotechnology Research Center, Southwest University, Beibei, Chongqing, China |
| Strain, strain background (*Agrobacterium tumefaciens*) | GV3101 | Lab stock | | *Arabidopsis* transformation, Biotechnology Research Center, Southwest University, Beibei, Chongqing, China |
| Strain, strain background (*Agrobacterium tumefaciens*) | LBA4404 | Lab stock | | Cotton transformation, Biotechnology Research Center, Southwest University, Beibei, Chongqing, China |
| Strain, strain background (*Escherichia coli* strain) | DH5α | Takara | Cat#9057 | DNA manipulations |
| Strain, strain background (*Beauveria bassiana*) | *B. bassiana* wild-type strain | China General Microbiologic-al Culture Collection Center (CGMCC) | Cat#CGMCC7.34 | Random insertion (T-DNA) library construction |
| Strain, strain background (*Verticillium dahliae*) | V991 | A gift of Prof. Guiliang Jian | | Analysis of plant disease resistance |
| Strain, strain background (*Verticillium dahliae*) | *V. dahliae* race L2-1 | A gift of Prof. Zhiying Ma | | Analysis of plant disease resistance |
| Strain, strain background (*Fusarium oxysporum*) | *F. oxysporum* f. sp. *vasinfectum* race 7 | A gift of Prof. Longfu Zhu | | Analysis of plant disease resistance |
| Strain, strain background (*Metarhizium anisopliae*) | *M. anisopliae* wild-type strain | A gift of Prof. Weiguo Fang | | Inhibition ring assay |
| Strain, strain background (*Botrytis cinerea*) | *B. cinerea* wild-type strain | CGMCC | Cat#3.4584 | Inhibition ring assay |
| Strain, strain background (*Alternaria brassicae*) | *A. brassicae* wild-type strain | CGMCC | Cat#3.7804 | Inhibition ring assay |
| Strain, strain background (*Alternaria brassicicola*) | *A. brassicicola* wild-type strain | CGMCC | Cat#3.7805 | Inhibition ring assay |

*Appendix 1 Continued on next page*

*Appendix 1 Continued*

| Reagent type (species) or resource | Designation | Source or reference | Identifiers | Additional information |
|---|---|---|---|---|
| Strain, strain background (*Aspergillus nidulans*) | *A. nidulans* wild-type strain | CGMCC | Cat#3.15737 | Inhibition ring assay |
| Strain, strain background (*Alternaria solani*) | *A. solani* wild-type strain | Lab stock | | Inhibition ring assay, Biotechnology Research Center, Southwest University, Beibei, Chongqing, China |
| Strain, strain background (*Arabidopsis thaliana*) | Col-0 | *Arabidopsis* Biological Resource Center | Cat#CS60000 | Analysis of plant disease resistance |
| Strain, strain background (*Gossypium hirsutum*) | *G. hirsutum, cv.* Jimian 14 | A gift of Prof. Zhiying Ma | | Analysis of plant disease resistance |
| Antibody | Anti-RFP (mouse monoclonal) | Thermo Scientific | Cat#MA5-15257 | WB (1:1000) |
| Antibody | Anti-ubiquitin (mouse monoclonal) | PTM-BIO | Cat#PTM-1107 | WB (1:2000) |
| Recombinant DNA reagent | pK2-*BbCRPA*5'-*Bar*-*BbCRPA*3' (plasmid) | This paper | | Deletion of *BbCAPA*, Biotechnology Research Center, Southwest University, Beibei, Chongqing, China |
| Recombinant DNA reagent | pK2-*Sur*-*BbCRPA* (plasmid) | This paper | | *BbCPRA* complementation, Biotechnology Research Center, Southwest University, Beibei, Chongqing, China |
| Recombinant DNA reagent | pK2-*HygB*-*BbCRPA* | This paper | | Expression of *BbCRPA* in *V. dahliae*, Biotechnology Research Center, Southwest University, Beibei, Chongqing, China |
| Recombinant DNA reagent | PLGN-*BbCRPA* (plasmid) | This paper | | Expression of *BbCRPA* in plants, Biotechnology Research Center, Southwest University, Beibei, Chongqing, China |
| Recombinant DNA reagent | pK2-*Sur*-BFL (plasmid) | This paper | | Functional verification of BbCrpa, Biotechnology Research Center, Southwest University, Beibei, Chongqing, China |
| Recombinant DNA reagent | pK2-*Sur*-$\Delta B^N$ (plasmid) | This paper | | Functional verification of N-terminus of BbCrpa, Biotechnology Research Center, Southwest University, Beibei, Chongqing, China |

*Appendix 1 Continued on next page*

*Appendix 1 Continued*

| Reagent type (species) or resource | Designation | Source or reference | Identifiers | Additional information |
|---|---|---|---|---|
| Recombinant DNA reagent | pK2-*Sur*-ΔB$^C$ (plasmid) | This paper | | Functional verification of C-terminus of BbCrpa, Biotechnology Research Center, Southwest University, Beibei, Chongqing, China |
| Recombinant DNA reagent | pK2-*Sur*-ΔB$^{C1325-1359}$ (plasmid) | This paper | | Functional verification of C-terminus of BbCrpa, Biotechnology Research Center, Southwest University, Beibei, Chongqing, China |
| Recombinant DNA reagent | pK2-*Sur*-ΔB$^{C1326-1359}$ (plasmid) | This paper | | Functional verification of C-terminus of BbCrpa, Biotechnology Research Center, Southwest University, Beibei, Chongqing, China |
| Recombinant DNA reagent | pK2-*Sur*-Y1325A (plasmid) | This paper | | Functional verification of Y1325 of BbCrpa, Biotechnology Research Center, Southwest University, Beibei, Chongqing, China |
| Recombinant DNA reagent | pK2-*Sur*-Y1325A-Y1341A (plasmid) | This paper | | Functional verification of Y1325-Y1341 of BbCrpa, Biotechnology Research Center, Southwest University, Beibei, Chongqing, China |
| Recombinant DNA reagent | pK2-*Sur*-Y1325A-Y1350A (plasmid) | This paper | | Functional verification of Y1325-Y1350 of BbCrpa, Biotechnology Research Center, Southwest University, Beibei, Chongqing, China |
| Recombinant DNA reagent | pK2-*Sur*-Y1341A-Y1350A (plasmid) | This paper | | Functional verification of Y1341-Y1350 of BbCrpa, Biotechnology Research Center, Southwest University, Beibei, Chongqing, China |
| Recombinant DNA reagent | pK2-*Sur*-Y1325A-Y1341A-Y1350A (plasmid) | This paper | | Functional verification of Y1325-Y1341-Y1350 of BbCrpa, Biotechnology Research Center, Southwest University, Beibei, Chongqing, China |
| Recombinant DNA reagent | pK2-*Sur*-DFL (plasmid) | This paper | | Functional verification of Drs2p, Biotechnology Research Center, Southwest University, Beibei, Chongqing, China |

*Appendix 1 Continued on next page*

*Appendix 1 Continued*

| Reagent type (species) or resource | Designation | Source or reference | Identifiers | Additional information |
|---|---|---|---|---|
| Recombinant DNA reagent | pK2-*Sur*-B$^N$-D$^{TMD}$-D$^C$ (plasmid) | This paper | | Functional verification of N-terminus of BbCrpa, Biotechnology Research Center, Southwest University, Beibei, Chongqing, China |
| Recombinant DNA reagent | pK2-*Sur*-D$^N$-D$^{TMD}$-B$^C$ (plasmid) | This paper | | Functional verification of C-terminus of BbCrpa, Biotechnology Research Center, Southwest University, Beibei, Chongqing, China |
| Recombinant DNA reagent | pK2-*Sur*-B$^N$-D$^{TMD}$-B$^C$ (plasmid) | This paper | | Functional verification of transmembrane domains of BbCrpa, Biotechnology Research Center, Southwest University, Beibei, Chongqing, China |
| Recombinant DNA reagent | pK2-*Sur*-D614R (plasmid) | This paper | | Functional verification of D614 of BbCrpa, Biotechnology Research Center, Southwest University, Beibei, Chongqing, China |
| Recombinant DNA reagent | pK2-*Sur*-I562E (plasmid) | This paper | | Functional verification of I562 of BbCrpa, Biotechnology Research Center, Southwest University, Beibei, Chongqing, China |
| Recombinant DNA reagent | eGFP::BbCrpa/BFL (plasmid) | This paper | | Localization of BbCrpa, Biotechnology Research Center, Southwest University, Beibei, Chongqing, China |
| Recombinant DNA reagent | eGFP::dB$^N$ (plasmid) | This paper | | Functional verification of N-terminus of BbCrpa, Biotechnology Research Center, Southwest University, Beibei, Chongqing, China |
| Recombinant DNA reagent | eGFP::dB$^C$ (plasmid) | This paper | | Functional verification of C-terminus of BbCrpa, Biotechnology Research Center, Southwest University, Beibei, Chongqing, China |
| Recombinant DNA reagent | B$^{N1-268}$::mRFP (plasmid) | This paper | | Functional verification of N-terminus of BbCrpa, Biotechnology Research Center, Southwest University, Beibei, Chongqing, China |

*Appendix 1 Continued on next page*

*Appendix 1 Continued*

| Reagent type (species) or resource | Designation | Source or reference | Identifiers | Additional information |
|---|---|---|---|---|
| Recombinant DNA reagent | B$^{N258-268}$::mRFP (plasmid) | This paper | | Functional verification of N-terminus of BbCrpa, Biotechnology Research Center, Southwest University, Beibei, Chongqing, China |
| Recombinant DNA reagent | B$^{N258-268 (K264A)}$::mRFP (plasmid) | This paper | | Functional verification of N-terminus of BbCrpa, Biotechnology Research Center, Southwest University, Beibei, Chongqing, China |
| Recombinant DNA reagent | B$^{N259-268}$::mRFP (plasmid) | This paper | | Functional verification of N-terminus of BbCrpa, Biotechnology Research Center, Southwest University, Beibei, Chongqing, China |
| Recombinant DNA reagent | B$^{N0}$::mRFP (plasmid) | This paper | | Functional verification of N-terminus of BbCrpa, Biotechnology Research Center, Southwest University, Beibei, Chongqing, China |
| Recombinant DNA reagent | mRPF::BbCrpa (plasmid) | This paper | | Localization of BbCrpa, Biotechnology Research Center, Southwest University, Beibei, Chongqing, China |
| Recombinant DNA reagent | mRFP::PH$^{OSBP}$ (plasmid) | A gift of Dr. Miguel A. Peñalva | | Labelling TGN, Biotechnology Research Center, Southwest University, Beibei, Chongqing, China |
| Recombinant DNA reagent | mRFP::BbRab5 (plasmid) | This paper | | Labelling early endosomes, Biotechnology Research Center, Southwest University, Beibei, Chongqing, China |
| Recombinant DNA reagent | mRFP::BbRab7 (plasmid) | This paper | | Labelling late endosomes, Biotechnology Research Center, Southwest University, Beibei, Chongqing, China |
| Recombinant DNA reagent | eGFP::Drs2p (plasmid) | This paper | | Localization of Drs2p, Biotechnology Research Center, Southwest University, Beibei, Chongqing, China |
| Chemical compound, drug | Cyclosporine A (CsA) | APEXBIO | Cat#B1922 | |
| Chemical compound, drug | Tacrolimus (FK506) | APEXBIO | Cat#B2143 | |
| Chemical compound, drug | Cinnamyl acetate (CIA) | Sigma-Aldrich | Cat#166170 | |

*Appendix 1 Continued on next page*

*Appendix 1 Continued*

| Reagent type (species) or resource | Designation | Source or reference | Identifiers | Additional information |
|---|---|---|---|---|
| Chemical compound, drug | Phosphinothricin | PhytoTech | Cat#P679 | |
| Chemical compound, drug | Chlorimuron-ethyl | Alfa Aesar | Cat#J66605 | |
| Commercial assay, kit | Fungal protein extraction kit | BestBio | Cat#BB-3136 | |
| Commercial assay, kit | PrimeScript RT reagent Kit | Takara | Cat#RR037A | cDNA synthesis |
| Commercial assay, kit | iQSYBR Green Supermix | Bio-Rad | Cat#1708882 | qRT-PCR |
| Software, algorithm | GraphPad Prism | GraphPad Software | https://www.graphpad.com | |
| Software, algorithm | ImageJ | ImageJ | http://rsb.info.nih.gov/ij/ | |
| Software, algorithm | CFX96 Real-Time System | Bio-Rad | | |
| Software, algorithm | FV10-ASW 3.0 Viewer | Olympus | | |
| Other | FM4-64 | Thermo Scientific | Cat#F34653 | Membrane-binding fluorescent dye |
| Other | FM1-43 | Thermo Scientific | Cat#T35356 | Membrane probe |

