## [Editor Report]

This important study reveals a new strategy for protecting plants from certain fungal pathogens. The authors show convincing data for a mechanism of resistance that involves sequestration of a fungal toxin into vacuoles. This work will be of interest to mycologists and scientists interested in plant biotechnology.

---

## [Decision Letter]

**Decision letter after peer review:**

Thank you for submitting your article "Cell detoxification of secondary metabolites by P4-ATPase mediated vesicle transport" for consideration by *eLife*. Your article has been reviewed by 3 peer reviewers, one of whom is a member of our Board of Reviewing Editors, and the evaluation has been overseen by Jürgen Kleine-Vehn as the Senior Editor. The reviewers have opted to remain anonymous.

Essential revisions:

While the reviewers were enthusiastic about this manuscript, they also raised a number of technical issues that need to be addressed before we can consider this manuscript further. To help the authors, if they choose to revise and re-submit to *eLife*, we have prepared the following summary of issues that must be addressed. In some cases, this may require additional experimentation. In others, the authors may be able to satisfy the concerns with edits to the text and a 'response to the reviewers' document to be included with a re-submission. The reviewers have provided a number of additional suggestions that we encourage the authors to consider to further improve their manuscript.

1. Please revise the text to clarify why the authors predicted that BbCrpa would function in plants, against CA, and to avoid overstating the broad applicability and strength of resistance (or include additional data).

2. Please include additional data to support the conclusions surrounding the location of protein and toxin. As is, the reviewers were concerned about cleaved GFP being misleading and early and late endosome distinctions (please see detailed comments from reviewers).

3. Please include a more detailed explanation/description of the transgenic lines used in this study.

4. All other points raised by the reviewers need to be addressed.

*Reviewer #1 (Recommendations for the authors):*

This reviewer had a hard time following the logic of why the authors expected that BbCrpa would work in plants. Because this was not explained in detail and the in planta data was not as robust as the rest of the paper, this reviewer views this result with excitement but skepticism.

*Reviewer #2 (Recommendations for the authors):*

Line 81: I would say "Most lipid ABC transporters".

*Reviewer #3 (Recommendations for the authors):*

The manuscript could be further strengthened by the authors by addressing the following issues.

1. The authors state that P4 ATPase is a phospholipid binding enzyme that acts as a flippase. However, the authors provide no data supporting this statement for the BbCrpa enzyme. Using phospholipid-protein overlay assays, they should determine which phospholipids bind to this enzyme and provide evidence that this enzyme pumps specific phospholipid substrates from the exofacial to the cytosolic leaflet of membranes.

2. The authors should consider using specific pharmacological endocytosis inhibitors to provide further confirmation of the movement of this enzyme from Golgi to vacuole.

3. The comparison of the amino acid sequences of the Crpa enzyme from closely related Bauveria barbiana and B. nivea could provide valuable insight into what makes B. barbiana resistant and B. nivea susceptible to CsA and FK506.

4. The authors should provide clear evidence for the expression of the BbCrpa gene at the protein level in transgenic Arabidopsis and cotton lines.

5. The authors should clearly state whether homozygous transgenic cotton and Arabidopsis lines were used in the V. dahliae infection assays.

6. In Figure 4 supplement 1F, mRFP and eGFP labels are interchanged.

7. Figure 5 supplement 1 Southern blot figures C and F are of poor quality.

8. The manuscript needs a careful English grammar review.

---

## [Author Response]

Essential revisions:While the reviewers were enthusiastic about this manuscript, they also raised a number of technical issues that need to be addressed before we can consider this manuscript further. To help the authors, if they choose to revise and re-submit to eLife, we have prepared the following summary of issues that must be addressed. In some cases, this may require additional experimentation. In others, the authors may be able to satisfy the concerns with edits to the text and a 'response to the reviewers' document to be included with a re-submission. The reviewers have provided a number of additional suggestions that we encourage the authors to consider to further improve their manuscript.1. Please revise the text to clarify why the authors predicted that BbCrpa would function in plants, against CA, and to avoid overstating the broad applicability and strength of resistance (or include additional data).

Thanks for your comments! CsA, a neutral lipophilic cyclic polypeptide, and FK506, a macrolide lactone, are all lipophilic compounds. We demonstrated that BbCrpa can confer the resistance of *B. bassiana* to CsA and FK506 *via* a vesicle-mediated transport pathway that targets the compounds into vacuoles for detoxification. We therefore speculated that cargoes transported by BbCrpa-associated vesicle trafficking for the detoxification may be a group of molecules with similar physicochemical or biological properties. Cinnamyl acetate (CIA) is a lipophilic toxin produced by phytopathogenic fungus *V. dahliae*. The common feature between CsA, FK506, and CIA is that they are all small lipophilic compounds. The cross-species resistance endowed by BbCrpa against CsA in *V. dahliae*, the common feature of the three toxins, as well as the successful application of herbicidal gene and insecticidal genes from bacteria in genetically modified crops, encouraged us to test whether BbCrpa can be exploited for detoxification of the fungal mycotoxin CIA in plants. Our next study will focus on what properties of cargos of BbCrpa-associated trafficking should have and how the system recognize the cargos.

According to the suggestion, we added words to describe why we predicted that BbCrpa would function in plants against CIA. In addition, based on the suggestions, to verify BbCrpa would function in plants, we used another two plant pathogens, *V. dahliae* race L2-1 and *Fusarium oxysporum* f. sp. *vasinfectum* race 7, to challenge *BbCRPA* transgenic Arabidopsis and cotton. The expression of *BbCRPA* in Arabidopsis and cotton significantly increased the resistance to the two phytopathogens. New data were show in Figure 5—figure supplement 1G-J.

2. Please include additional data to support the conclusions surrounding the location of protein and toxin. As is, the reviewers were concerned about cleaved GFP being misleading and early and late endosome distinctions (please see detailed comments from reviewers).

Thanks for your comments! In order to rule out the interference of eGFP/mRFP/5-FAM cleaved from the fused targets, we added eGFP, mRFP, and 5-FAM as controls, and compared the localizations of eGFP/mRFP with that of eGFP/mRFP::BbCrpa, and 5-FAM with 5-FAM-labeled CsA in *B. bassiana*. Our data indicated that the distribution of eGFP/mRFP or 5-FAM alone differ from that of eGFP/mRFP fused proteins or 5-FAM labeled CsA. We added Figure 2—figure supplement 2 to show the difference. It can be seen that eGFP/mRFP signal appeared in the cytosol and no significant signal was found in vacuoles (A and C); in contrast, eGFP/mRFP::BbCrpa signal acuminated in vacuoles (B and D). Similarly, strong 5-FAM signal was found in the vacuoles in Δ*BbCRPA* cells (E), while almost no 5-FAM-CsA was found in the vacuoles (F). Our observation indicated that the localization of eGFP/mRFP and 5-FAM alone is different from that of the eGFP/mRFP-fused proteins and 5-FAM-labelled CsA.

3. Please include a more detailed explanation/description of the transgenic lines used in this study.

Thanks for your comments! According to the suggestion, we added the detailed explanation/description of how transgenic lines were selected and verified (lines 316-324). Plant transformation and the transgenic line generation were detailed in the section of Methods and Materials (lines 634-646). In addition, flanking sequencing information of a transgenic cotton line was shown in the Figure 5—figure supplement 1F that indicates the *bona fide* location of the transgene in cotton chromosome.

4. All other points raised by the reviewers need to be addressed.

Our point-by-point responses to reviewers are in following.

Reviewer #1 (Recommendations for the authors):This reviewer had a hard time following the logic of why the authors expected that BbCrpa would work in plants. Because this was not explained in detail and the in planta data was not as robust as the rest of the paper, this reviewer views this result with excitement but skepticism.

Thanks for your comments! The common lipophilic feature of CsA, FK506, and CIA, and the cross-species resistance of BbCrpa to CsA in *V. dahliae* encouraged us to test whether BbCrpa can be exploited in plants. According to your suggestion, we added the description about why we expressed the fugal gene in plants (lines 310-313) and how transgenic plants were generated (lines 634-646), as well as the data of challenging with more pathogens (Figure 5—figure supplement 1G-L).

Reviewer #2 (Recommendations for the authors):Line 81: I would say "Most lipid ABC transporters".

Thanks! Following the suggestion, we have changed "Most ABC transporters" to "Most lipid ABC transporters". Please see line 81.

Reviewer #3 (Recommendations for the authors):The manuscript could be further strengthened by the authors by addressing the following issues.1. The authors state that P4 ATPase is a phospholipid binding enzyme that acts as a flippase. However, the authors provide no data supporting this statement for the BbCrpa enzyme. Using phospholipid-protein overlay assays, they should determine which phospholipids bind to this enzyme and provide evidence that this enzyme pumps specific phospholipid substrates from the exofacial to the cytosolic leaflet of membranes.

Thanks for your suggestion! P4-ATPases are known as flippases that can catalyze the translocation of aminophospholipids, e.g. phosphatidylethanolamine (PE) and phosphatidylserine (PS), from the plasma membrane to the inner membrane system. To determine the substrates of P4-ATPase BbCrpa, we observed the distribution of phosphatidylethanolamine (PE) and phosphatidylserine (PS indicated by Lact-C2) in wild-type *B. bassiana* and Δ*BbCRPA*. The results showed that PE was mainly distributed on the cytosolic face in the wild-type, while no significant PE signal was found in the cytosol of Δ*BbCRPA.* For PS, when the gene was knocked out, PS trafficking from membrane to inner compartments was interrupted, thus resulting accumulation of PS in the plasma membrane. These data indicated that BbCrpa participates in transporting of PE and PS. New data were shown in Figure 1—figure supplement 3A and B.

2. The authors should consider using specific pharmacological endocytosis inhibitors to provide further confirmation of the movement of this enzyme from Golgi to vacuole.

Thanks for your suggestion! Brefeldin A (BFA) that disrupts Golgi structure and inhibits Golgi function is used as a endocytosis inhibitor (João Conde et al., Self-assembled RNA-triple-helix hydrogel scaffold for microRNA modulation in the tumour microenvironment, *Nature Material,* 2015 | DOI: 10.1038/NMAT4497) or a exocytosis inhibitor ( Andrei I. Ivanov, Pharmacological inhibitors of exocytosis and endocytosis: novel bullets for old targets, *Methods in Molecular Biology*, 2014 | DOI 10.1007/978-1-4939-0944-5_1). We used BFA to treat *B. bassiana* and observed the movement of BbCrpa from Golgi to vacuole. The data showed that BFA treatment decreased BbCrpa accumulation in the vacuole. New data were added in Figure 3—figure supplement 2.

3. The comparison of the amino acid sequences of the Crpa enzyme from closely related Bauveria barbiana and B. nivea could provide valuable insight into what makes B. barbiana resistant and B. nivea susceptible to CsA and FK506.

That's a good idea! But we failed to found BbCrpa homologue in the genome of B*. nivea*. Thus, we are unable to compare the amino acid sequences of the Crpa with that from *B. nivea*.

4. The authors should provide clear evidence for the expression of the BbCrpa gene at the protein level in transgenic Arabidopsis and cotton lines.

Thanks for your comments! Yes, it is better to provide clear evidence for the expression of the BbCrpa gene at the protein level in transgenic plants. To this end, we had asked bio-companies to generate BbCrpa antibody. Unfortunately, the antibodies did not work well. We guess the failure is because of that BbCrpa is a membrane protein with ten transmembrane-spanning segments, the molecular of weight is 151.8 kDa. Although we failed to obtain BbCrpa data at protein level, our evidences on the expression of *BbCRPA* at transcriptional level and the defect of BbCrpa in *B. barbiana* that results in the decrease of the resistance to CsA/FK506, as well as the ectopic expression of the gene in *Verticilium* and in plants increased the resistance to CIA and CsA/FK506 respectively, can reflect the function of BbCrpa at the protein level.

5. The authors should clearly state whether homozygous transgenic cotton and Arabidopsis lines were used in the V. dahliae infection assays.

Thanks for your comments! We used homozygous transgenic cotton and Arabidopsis plants for resistance assay. According to your suggestion, we have added the more information about transgenic materials in the section of Methods and Materials. Please see lines 634-646.

6. In Figure 4 supplement 1F, mRFP and eGFP labels are interchanged.

Thanks for your reminder! The error had been corrected according to the suggestion.

7. Figure 5 supplement 1 Southern blot figures C and F are of poor quality.

According to your advice, we did the experiments again and replaced the original figure with new one (Figure 5—figure supplement 1E).

8. The manuscript needs a careful English grammar review.

Thanks for your suggestion! The manuscript had been edited carefully.